# Beliefs, compulsive behavior and reduced confidence in control

**Lionel Rigoux[1,2], Klaas E. Stephan[1,2], Frederike H. Petzschner** **[3,4,5]** *

**1** Max Planck Institute for Metabolism Research, Cologne, Germany, **2** Translational Neuromodeling Unit, Institute for Biomedical Engineering, University of Zurich and Swiss Federal Institute of Technology Zurich, Zurich, Switzerland, **3** Robert J. and Nancy D. Carney Institute for Brain Science, Brown University, Providence, Rhode Island, United States of America, **4** Department of Psychiatry and Human Behavior, Brown University, Providence, Rhode Island, United States of America, **5** Center for Digital Health, Brown University, Providence, Rhode Island, United States of America

* frederike_petzschner@brown.edu

**Data Availability Statement:** Code is available on github (https://github.com/lionel-rigoux/beliefs-compulsions-and-reduced-confidence-in-control) and cited in the paper.

## Abstract

OCD has been conceptualized as a disorder arising from dysfunctional beliefs, such as overestimating threats or pathological doubts. Yet, how these beliefs lead to compulsions and obsessions remains unclear. Here, we develop a computational model to examine the specific beliefs that trigger and sustain compulsive behavior in a simple symptom-provoking scenario. Our results demonstrate that a single belief disturbance–a lack of confidence in the effectiveness of one's preventive (harm-avoiding) actions–can trigger and maintain compulsions and is directly linked to compulsion severity. This distrust can further explain a number of seemingly unrelated phenomena in OCD, including the role of not-just-right feelings, the link to intolerance to uncertainty, perfectionism, and overestimation of threat, and deficits in reversal and state learning. Our simulations shed new light on which underlying beliefs drive compulsive behavior and highlight the important role of perceived ability to exert control for OCD.

## Author summary

Obsessive-Compulsive Disorder (OCD) remains a perplexing condition for both scientists and healthcare professionals. A common theory posits that compulsive behaviors in OCD patients stem from dysfunctional beliefs, such as doubting their own actions and perceptions or harboring an exaggerated fear of potential dangers. Experimentally probing whether and how these beliefs lead to compulsions, however, is challenging because we cannot easily isolate and manipulate individual beliefs.

Here, we used an innovative approach to study the connection between beliefs and compulsions. We created a large set of simulations of many different individuals acting under different beliefs, to determine which of these beliefs trigger compulsions. We find that compulsive behavior primarily originates from a single belief: a lack of trust in one's ability to prevent harm effectively. This belief not only explains the emergence of compulsive behaviors but also why they persist.

**Funding:** This work was supported by the René and Susanne Braginsky Foundation (KES), the ETH Foundation (KES), the University of Zurich (KES), and the Brainstorm Program at the Robert J. & Nancy D. Carney Institute for Brain Science (FHP). The funders had no role in study design, data collection and analysis, decision to publish, or preparation of the manuscript.

Our simulations also provide insights into other aspects of OCD, including connections to beliefs about danger and perfectionism, as well as learning deficits observed in patients. By highlighting the crucial role of perceived control, our work contributes significantly to the understanding of OCD and opens new avenues for studying this intricate disorder.

## Introduction

Obsessive-compulsive disorder (OCD) is an enigmatic disorder that has inspired a range of physiological and cognitive theories, with different postulated causes and failure modes [1–7]. So far, a consensus on its origins and mechanisms has not been reached [8]. OCD is characterized by two core symptoms: obsessions and compulsions. Obsessions are recurrent and persistent thoughts, urges, or images that are intrusive and unwanted. Compulsions are repetitive behaviors or mental acts that patients feel compelled to do. They are often thought to be initiated as a counter-action in response to an obsession [9]. For example, a patient obsessing about fears of contaminations may develop a compulsion to excessively wash her hands.

Cognitive-behavioral theories have proposed that OCD is driven by dysfunctional beliefs [1,3,4,10–12], including an overestimation of threat, inflated sense of responsibility, need to control thoughts, perfectionism, and intolerance to uncertainty or pathological doubt [13]. The importance of these beliefs for the diagnosis of OCD has since been recognized by the Diagnostic Statistical Manual (DSM-5) where they are listed under the specifications of the disorder: *"Many individuals with obsessive-compulsive disorder (OCD) have dysfunctional beliefs. These beliefs can include an inflated sense of responsibility and the tendency to overestimate threat; perfectionism and intolerance of uncertainty; and over-importance of thoughts (e.g., believing that having a forbidden thought is as bad as acting on it) and the need to control thoughts."*[9].

Still, it remains unclear how and if these beliefs lead to obsessions and compulsions. For instance, the mere existence of a dysfunctional belief does not necessarily mean it *causes* the symptom. Instead, such a belief could also arise in response to experiencing the symptom. For example, while pathological doubt could cause excessive checking behavior [12], forced excessive checking and memory manipulations also induce doubts in healthy control groups [14–16]. Similarly, it is unclear why the overestimation of threat seems to specifically trigger compulsive avoidance responses in OCD patients, but not in patients with general anxiety disorder or panic disorder [17,18]. In sum, in OCD, it is not clear which beliefs and behaviors are causes and which are consequences—and whether this relation is consistent across patients [8].

Identifying aberrant beliefs that are drivers of compulsive symptoms would have substantial benefits: therapies could target core cognitive deficits more precisely, and experimental assessments could be devised to obtain biomarkers. Unfortunately, due to the circularity of the problem, disentangling primary cognitive or behavioral alterations from secondary impairments is difficult [19]. Empirically this would require a range of experiments that can selectively manipulate every single belief in patients and meticulously measure every resultant change in behavior. One promising alternative strategy to address this challenge is to adopt a computational approach for testing mechanistic hypotheses about the emergence of symptoms and signs in mental disorders [19–23]. For instance, mathematical models of cognition allow one to simulate the behavior of many agents who act under entirely different belief sets. They may thus help identify which beliefs are necessary and sufficient to elicit a particular behavior.

Here, we use such a computational approach to identify the relationships between compulsions and beliefs. First, we describe a simple prototypical symptom-provoking scenario (hand-washing to avoid contamination) with a parsimonious model of the phenomenon of interest ('minimal model' [24,25]) (Fig 1, Materials and methods–Computational Model). We then analyzed how agents that act under entirely different beliefs would behave in this scenario, in order to clarify who develops compulsions and how these compulsions relate to an individual's belief set. By simulating a variety of belief systems and observing the resultant behaviors in controlled computational models, we can more accurately discern the dynamic, non-trivial interplay between beliefs, compulsions, and changing environments, to systematically determine which beliefs are necessary and sufficient to trigger specific compulsive behaviors.

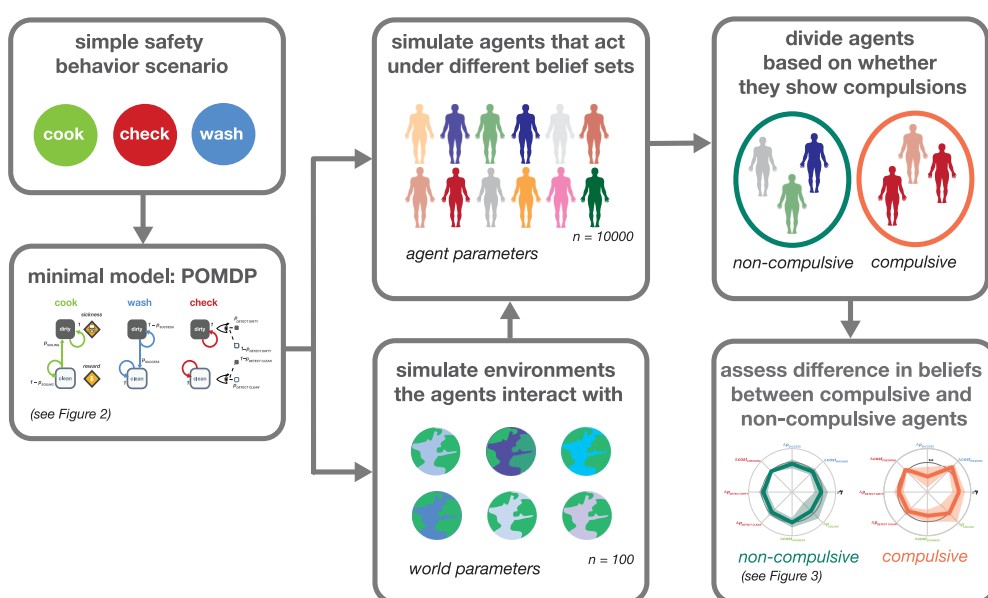

**Fig 1. A Schematic step-by-step example of our minimal model approach.** First, a simple symptom-provoking scenario is described by a parsimonious computational model (minimal model). Then the behavior of individual actors under this model is simulated to find subgroups with and without symptomatic behavior, respectively. Next, the difference in parameter values between these groups is assessed in order to discover symptom-relevant model parameters. **B** The concrete minimal model approach used in this paper. We use a simple safety behavior scenario—handwashing—modeled by a Partially Observable Markov Decision Process (POMDP) to simulate agents that interact with different environments (defined by nine world parameters) under different individual belief sets (defined by nine agent parameters = subjective beliefs about the world parameters). We then divide the agents into two groups, those that act compulsively (repeated washing and checking, compulsive group) and those that do not (non-compulsive group). Next, we compare the groups to assess which beliefs are necessary and sufficient to create compulsive behavioral patterns. See Materials and Methods –Computational Model and Simulations for a detailed description of each step.

In our computational model, we use the scenario of cooking dinner for guests as an illustrative example to explore the mechanisms of compulsive behaviors. It's essential to understand that the act of cooking here is not the focus, but rather a relatable example, serving as a proxy for any activity or time window where there's a potential for harm or damage. Our intention is to provide a straightforward and intuitive scenario, demonstrating how the model can be extrapolated to a wide array of activities where concerns about harm are relevant.

In this specific scenario, agents are preoccupied with the possibility of transmitting germs from their hands to the food, which could potentially result in illness for their guests (Fig 2). This concern requires them to constantly reassess their situation based on their actions and observations: deciding whether it's safe to cook given their current perception of hand cleanliness, or whether they should engage in behaviors like washing their hands or checking them for cleanliness. Agents are thus constantly updating their beliefs about whether the hands are dirty or clean.

Within this framework, several factors contribute to the potential development of compulsive behaviors. Foremost are the actual properties of the agent's environment, which we term 'world parameters.' These parameters include factors like the severity of potential contamination consequences and the likelihood of hands becoming contaminated in that environment. Additionally, the agents' beliefs about their environment, or 'agent parameters,' are equally influential. These beliefs, which may not always align with the actual risk in their environment, can significantly alter an agent's behavior (Materials and methods - Simulations).

Our simulations include a number of belief distortions, where agents' subjective beliefs diverge from the actual world parameters. These distortions encompass a range of dysfunctional beliefs commonly associated with OCD. For example, agents might overestimate the severity of contamination consequences or experience pathological doubts about their perception and actions' accuracy and effectiveness (Fig 2, see Table 1 for a full list of possible belief distortions). By simulating a large variety of world-agent combinations, we aim to identify those beliefs that lead to the emergence of compulsive behaviors (Fig 1).

This type of simulation allows us to address three pivotal questions: (i) What specific beliefs trigger compulsive behavior? (ii) How do these beliefs relate to the expression and severity of compulsive symptoms? (iii) How do these beliefs explain other phenomena of OCD?

## Results

### Differences between compulsive and non-compulsive agents

To explore the impact of false beliefs on the genesis of compulsions, we used a computational model (POMDP) to simulate the behavior of a large number of agents acting under different beliefs in a simple exemplary handwashing scenario (Fig 2, Materials and methods–Computational Model).

For our first set of analysis, we categorized these agents into two distinct groups: one comprising agents who exhibited compulsive behaviors in this scenario, specifically repetitive handwashing and/or checking (n = 5,000 agents), and another consisting of agents who did not display compulsive behaviors (n = 5,000 agents) (Fig 1B, see Materials and methods - Simulations). The purpose of this segregation was to meticulously evaluate which specific beliefs act as drivers of compulsive behavior.

**Compulsive agents underestimate the effectiveness of their avoidance action.**   First, we compared the individual belief distortions of the compulsive and non-compulsive group (Fig 3). While agent's beliefs could deviate substantially from the real world (quartiles indicated by shaded areas in Fig 3), on average what distinguished the compulsive and non-compulsive group was a single belief: Compulsive agents underestimated the effectiveness of their active

**A.** model of hand-washing

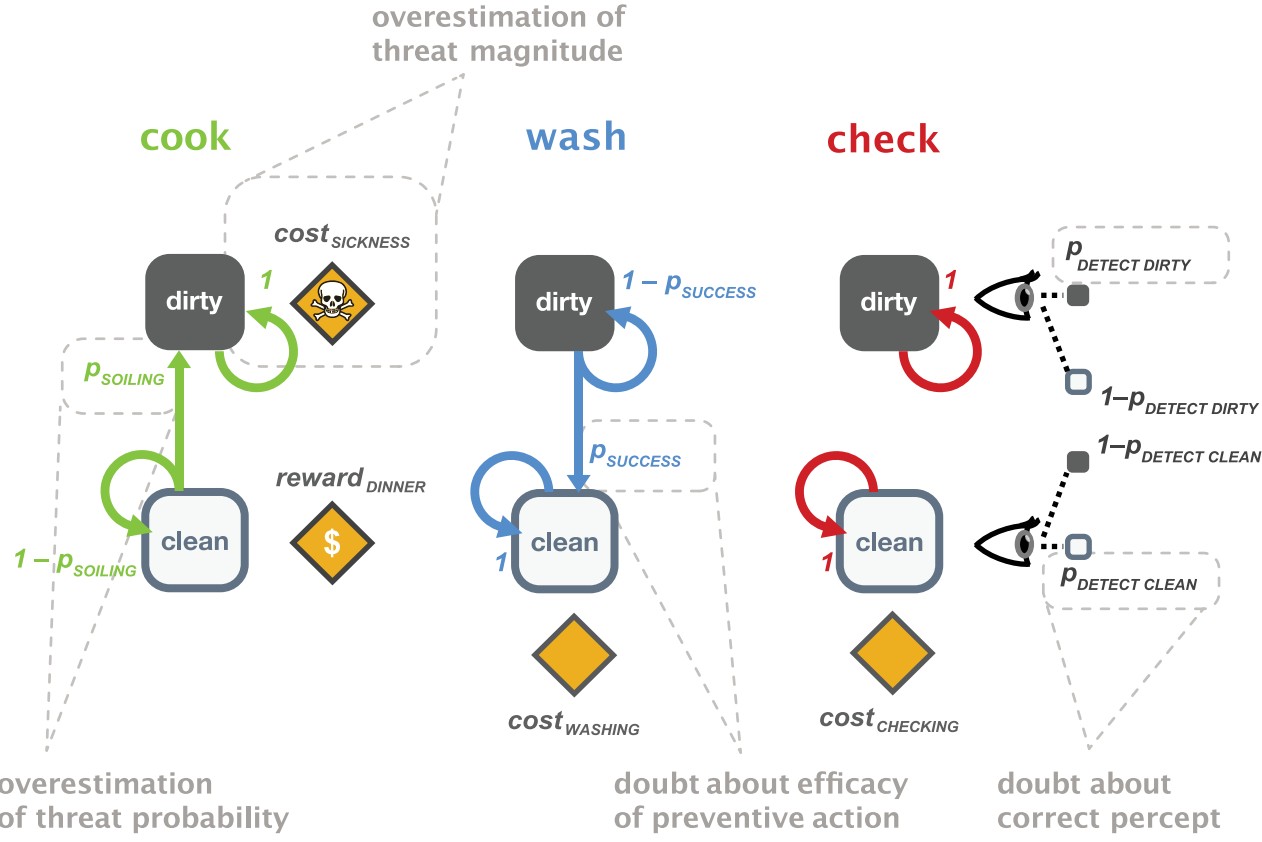

**B.** subjective optimal action sequence

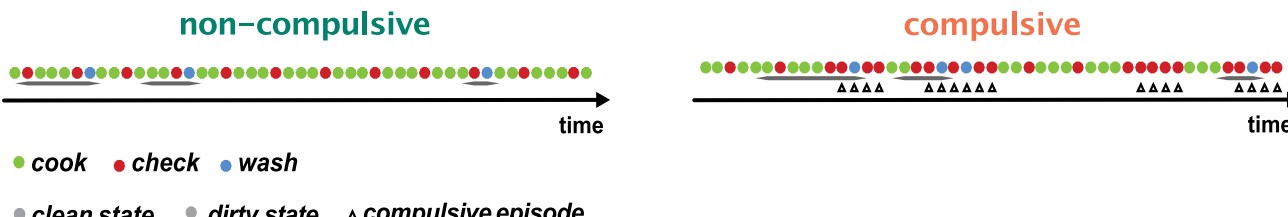

**Fig 2. A** Schematic representation of the handwashing scenario and the POMDP formalization. Our scenario considers agents who worry about contamination while cooking dinner for their guests. They know that germs from their hands may be transmitted to the food and cause their guests to become sick. In every iteration they have to choose between three actions (cooking, washing, or checking) while their hands can be in two possible states (dirty or clean). **Left:** If an agent decides to cook, two different outcomes are possible: If the hands are clean, cooking results in a pleasant meal with the guests (positive outcome: $reward_{DINNER}$). In contrast, if the hands are dirty during cooking, the food gets contaminated and causes sickness of the guests (negative outcome: $cost_{SICKNESS}$). Importantly, while cooking, clean hands can get dirty with probability $p_{SOILING}$ (state change from clean to dirty) or stay clean with probability $1 - p_{SOILING}$. **Middle:** To clean dirty hands, agents can decide to wash them. In this case, the hands will become clean with probability $p_{SUCCESS}$ or stay dirty with probability $1 - p_{SUCCESS}$. In case the hands were already clean, they will simply stay clean. **Right:** Finally, to learn more about the state of their hands, agents can also check. Checking itself does not change a state (no arrow between dirty and clean), but provides additional information about the state (observation indicated by the eye symbol). If the hands are dirty, the agent will correctly detect this state (through checking) with probability $p_{DETECT\ DIRTY}$. Conversely, if the hands are clean, the probability of correctly detecting this state is $p_{DETECT\ CLEAN}$. Agents gain no information about the state when they cook or wash. Thus, the observation or detection probability for these states is at chance level ($p = 0.5$). Both checking and washing are associated with a small cost (negative outcome: $cost_{WASHING}$ and $cost_{CHECKING}$, respectively) when executed, which represents the effort spent performing this action. Note, in the model we do not only describe the actual contingencies of the world, but also an individual's subjective belief about those contingencies. This allows us to capture discrepancies between beliefs and actual relationships in the world and how these false beliefs might result in repeated checking and washing. Several of these false beliefs directly relate to dysfunctional beliefs in OCD and are indicated in light gray (see also Table 1). Symbols: Squares represent states, arrows denote transitions

due to actions, eyes indicate information gained from an action, and diamonds the outcomes (see Material and Methods–Computational Model for a detailed mathematical description). **B** Typical action sequence of an agent in the compulsive and non-compulsive group. Compulsive episodes (multiple checks or washes in a row) are indicated by a triangle. Notably, compulsive episodes can be triggered even in the absence of objectively dirty hands (right plot: third compulsive episode).

avoidance action ($\Delta p_{SUCCESS}$ = difference in the believed and actual effectiveness of washing to get clean hands, *Cohen's d = 1.17*, bootstrapped *t*-test *p = 0.006*, see Table 2 for a comparison of all beliefs; Note that in order to avoid overinflated significance given our large sample size, all statistical tests were performed using a bootstrap method, see Material and Methods—Statistical Analyses; Also whenever we refer to avoidance action in this paper, we mean active avoidance (choosing an action to prevent a negative outcome) as opposed to passive avoidance.). Moreover, while non-compulsive agents on average had a veridical belief about the effectiveness of hand washing ($\Delta p_{SUCCESS}$ = −0.036(0.20), *p* = 0.409), compulsive agents systematically underestimated that effectiveness ($\Delta p_{SUCCESS}$ = -0.37 (0.32), *p* < 0.001).

**Compulsive and non-compulsive groups do not differ in any other beliefs.** Notably, a difference between the two groups could not be detected in any other beliefs that were proposed to drive compulsions, such as doubts about perception ($p_{DETECT,DIRTY}'$: *Cohen's d = 0.05, p = 0.500*; $p_{DETECT,CLEAN}'$: *Cohen's d = 0.02, p = 0.451*) or an overestimation of threat ($cost_{SICKNESS}'$: *Cohen's d = 0.87, p = 0.064*), although compulsive agents were significantly biased ($\Delta cost_{SICKNESS}$ = −0.151(0.215), *p* = 0.031; see Table 2). The same effects were also observed when the simulations were repeated under a different definition of compulsions that did not require checking (see Fig A and Table A in S1 Supplementary Material). This pattern suggests that the primary dysfunctional belief leading to compulsive behaviors in our simulated agents is a diminished confidence in successfully executing preventive actions, thereby exerting control over perceived threats. However, this should not be interpreted as excluding the possibility of other belief distortions arising as a consequence of or contributing to compulsive behaviors. While not necessary for the emergence of compulsions, these additional belief distortions may still influence the nature and severity of compulsive symptoms, a relationship we explore in greater detail later in our analysis.

**Table 1. Description of the world parameters and subjective belief about world parameter (agent parameters) and their proposed relationship with dysfunctional beliefs.** See also Fig 2 for a schematic representation of the parameter relationships. Δ params = agent params–world params.

| Parameter type | World parameter | Subjective belief | Description | Potential relation to dysfunctional beliefs in OCD |
|---|---|---|---|---|
| state transition probability | $p_{SOILING}$ | $p_{SOILING}'$ | probability of getting dirty hands while cooking | Overestimation of threat probability ($\Delta p_{SOILING} > 0$) |
| | $p_{SUCCESS}$ | $p_{SUCCESS}'$ | probability of successfully cleaning the hands when dirty | Doubt about action: Underestimation of effectiveness of preventive action ($\Delta p_{SUCCESS} < 0$) |
| outcome | $reward_{DINNER}$ | $reward_{DINNER}'$ | reward for a successful dinner | |
| | $cost_{SICKNESS}$ | $cost_{SICKNESS}'$ | cost of sickness when cooking with dirty hands | Overestimation of threat magnitude ($\Delta cost_{SICKNESS} < 0$) |
| | $cost_{WASH}$ | $cost_{WASH}'$ | cost of washing the hands | |
| | $cost_{CHECK}$ | $cost_{CHECK}'$ | cost of checking the state of the hands | |
| observation probability | $p_{DETECT\ DIRTY}$ | $p_{DETECT\ DIRTY}'$ | probability of detecting dirty hands via checking | Doubt about observation/percept: Underestimation of correct detection ($\Delta p_{DETECT\ DIRTY} < 0$) |
| | $p_{DETECT\ CLEAN}$ | $p_{DETECT\ CLEAN}'$ | probability of detecting clean hands via checking | Doubt about observation/percept: Underestimation of correct detection ($\Delta p_{DETECT\ CLEAN} < 0$) |
| discounting | $\gamma$ | $\gamma'$ | discount of future rewards | |

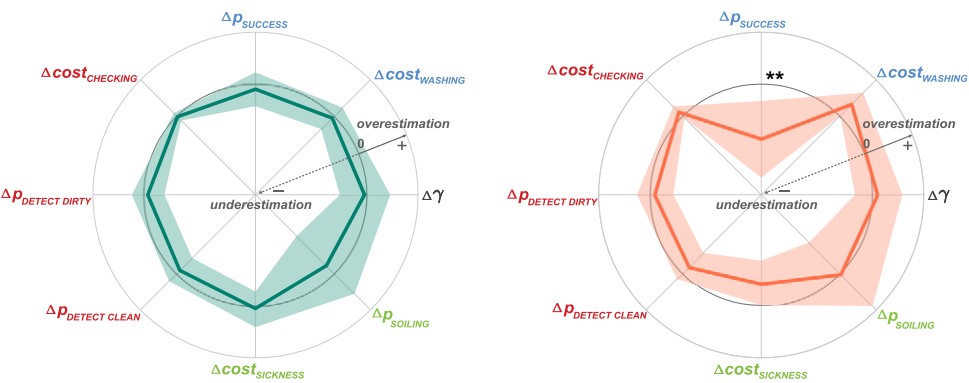

**Fig 3. Compulsive agents underestimate the probability of successful washing.** Belief distortion for the non-compulsive group (**A**, green) and compulsive group (**B**, orange) based on Simulation 1A. The solid-colored lines represent the average differences between agent and world parameters ($\Delta$ *params*: subjective beliefs–world parameters) across agents in each group, respectively. The inner light gray circle represents the zero line where world and agent parameters would be identical. This would represent beliefs that reflect a veridical representation of the world. Colored shades indicate the interval formed by the 1st and 3rd quartile of the group's belief. Deviations of the solid-colored lines towards the center of the circle indicate that agents believe the respective parameter is smaller than it actually is (underestimation, $\Delta$ *params*<0). Deviations away from the center, from the midline to the outer circle, indicate that agents overestimate that parameter (overestimation, $\Delta$ *params*>0). For most parameters, the compulsive and non-compulsive group have a veridical representation of the environment on average (average close to the midline). There is one notable difference between the two groups: Compulsive agents significantly underestimated the effectiveness of washing their hands compared to non-compulsive ones ($\Delta p_{SUCCESS}$<0; Group comparison: *Cohen's d = 1.17*, bootstrapped t-test p = 0.006). **: p < 0.01.

## Relationship between false beliefs and compulsion severity

So far, we found that compulsive agents differ from non-compulsive agents in a single belief: They underestimate the effectiveness of their preventive action (washing). Next, we assessed how this and other belief distortions affect the expression and severity of compulsions, including their duration, prevalence, and the number of compulsive episodes.

**Compulsion severity increases with distrust in avoidance effectiveness.** We found that a growing distrust in the effectiveness of avoidance behaviors correlates with an increased severity of compulsive episodes, as shown in Fig 4A. Specifically, the number, length, and frequency of compulsive episodes increased in proportion to the extent agents underestimated

**Table 2. Difference in world parameters and subjective beliefs (agent parameters),Δ params, for the compulsive and non-compulsive group in Simulation 1A (Mean (std)).** Statistical comparison against 0 (no belief distortion) and between compulsive and non-compulsive group: bootstrapped t-test, d = Cohen's d. * indicates significance.

| | Non-compulsive | | Compulsive | | Group difference | |
|---|---|---|---|---|---|---|
| *Parameter* | *Estimate* | *t-test (p-value)* | *Estimate* | *t-test (p-value)* | *d* | *t-test (p-value)* |
| $\Delta \gamma$ | -0.012 (0.263) | 0.496 | 0.035 (0.264) | 0.447 | -0.196 | 0.471 |
| $\Delta cost_{WASH}$ | -0.016 (0.165) | 0.468 | 0.112 (0.179) | 0.057 | -0.850 | 0.087 |
| $\Delta p_{SUCCESS}$ | **-0.036 (0.201)** | **0.409** | **-0.373 (0.321)** | **<0.001***** | **1.172** | **0.006**** |
| $\Delta cost_{CHECK}$ | -0.004 (0.055) | 0.484 | 0.039 (0.077) | 0.111 | -0.630 | 0.133 |
| $\Delta p_{DETECT\ DIRTY}$ | -0.021 (0.170) | 0.454 | -0.030 (0.178) | 0.421 | 0.049 | 0.500 |
| $\Delta p_{DETECT\ CLEAN}$ | -0.026 (0.178) | 0.438 | -0.060 (0.188) | 0.269 | 0.196 | 0.451 |
| $\Delta cost_{SICKNESS}$ | 0.020 (0.182) | 0.463 | -0.151 (0.215) | 0.031* | 0.868 | 0.064 |
| $\Delta p_{SOILING}$ | -0.071 (0.413) | 0.421 | 0.007 (0.432) | 0.496 | -0.188 | 0.463 |

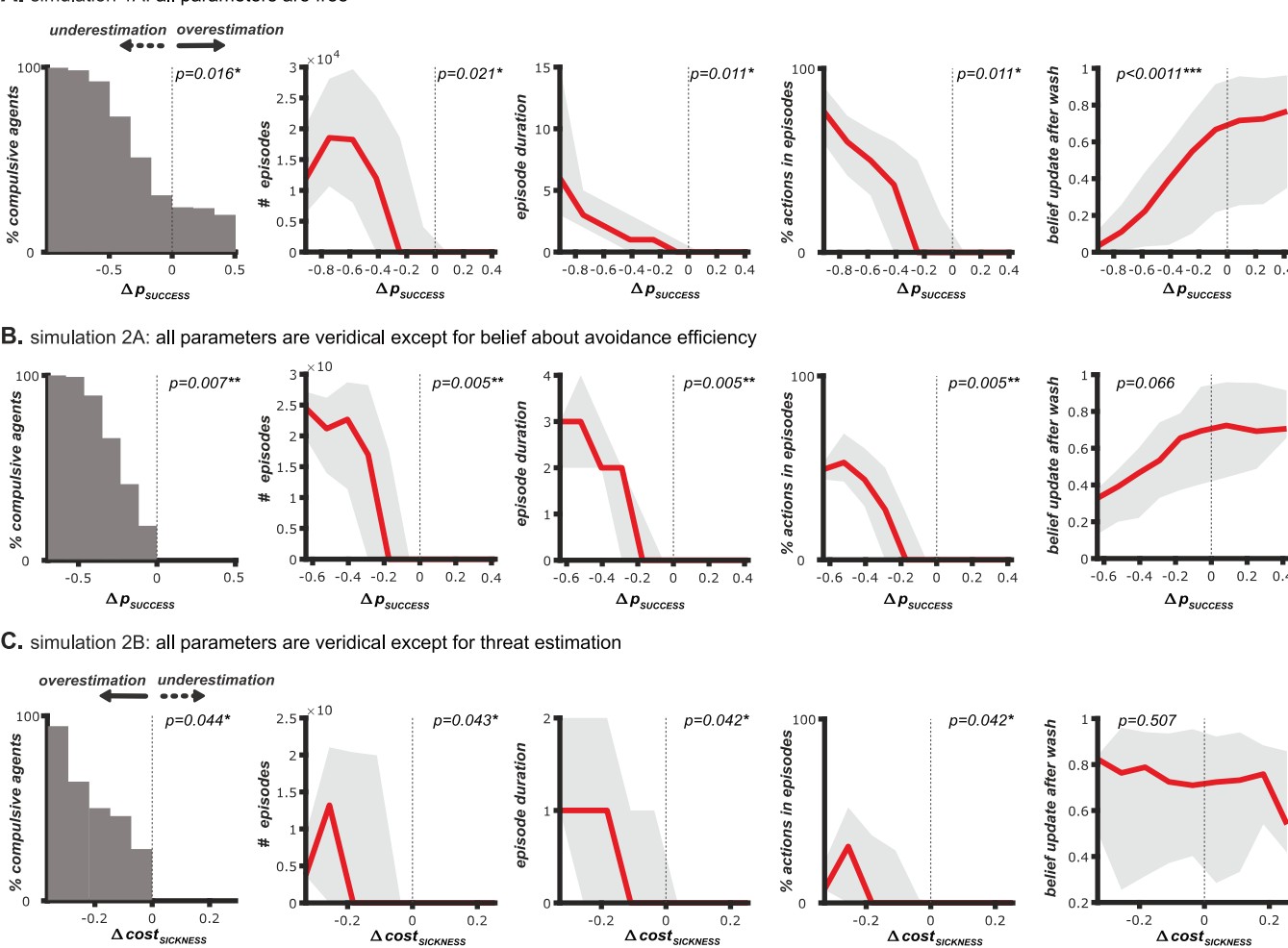

**Fig 4. Underestimation avoidance action effectiveness but not overestimation of threat increases compulsion severity. A.** Data for simulation 1A (all parameters variable). The x-axis on all plots shows the degree to which agents' beliefs deviated from the true effectiveness of washing, $\Delta p_{SUCCESS}$. Here, $\Delta p_{SUCCESS} < 0$ indicates an underestimation of washing effectiveness, $\Delta p_{SUCCESS} = 0$ a veridical representation, and $\Delta p_{SUCCESS} > 0$ an overestimation. From left to right: Relationship between $\Delta p_{SUCCESS}$ and the percentage of agents that show compulsive episodes (left), the number of compulsive episodes (center left), the duration of compulsive episodes (center), the percentage of actions that are part of a compulsive episodes as opposed to outside a compulsive episode (center-right), and the amount of belief update about the state (hand cleanliness) after a wash action (right). **B.** Same plots but for data for simulation 2A (all parameters veridical except belief about washing effectiveness) **C.** Relation between overestimation of threat and compulsion severity: Data for simulation 2B (all parameters veridical except belief about cost of sickness). The x-axis on all plot shows the degree to which agents deviate in their belief about cost of sickness, $\Delta cost_{SICKNESS}$. Whereby $\Delta cost_{SICKNESS} < 0$ indicates an overestimation of the negative cost of sickness, $\Delta cost_{SICKNESS} = 0$ a veridical representation and $\Delta cost_{SICKNESS} > 0$ an underestimation. From left to right: Relationship between $\Delta cost_{SICKNESS}$ and the percentage of agents that show compulsive episodes (left), the number of compulsive episodes (center left), the duration of compulsive episodes (center) the percentage of actions that are part of a compulsive episodes as opposed to outside a compulsive episode (center right), and amplitude of the belief update after washing. Statistics: bootstrapped Spearmann correlation. * indicates significance.

the effectiveness of the washing behavior. This trend was not observed for other belief distortions including overestimations of threat probability or magnitude, or doubts regarding observations (bootstrapped Spearman rank correlation of $\Delta p_{SUCCESS}$ with $\Delta cost_{SICKNESS}$, $\Delta p_{SOILING}$, $\Delta p_{DETECT\ CLEAN}$ or $\Delta p_{DETECT\ DIRTY}$; see Table B in S1 Supplementary Material).

Moreover, the number of compulsive episodes grew with an increase in the underestimation of washing success. There was a point where the number of compulsive episodes no longer increased further as the length of a single episode had become so long that multiple episodes merged into fewer long ones (Fig 4A). This was reflected in the proportion of actions taken

compulsively: For the largest underestimation of washing success, the large majority of actions was part of a compulsive episode (washing or checking), as opposed to outside a compulsive episode (cooking). Critically, if washing was believed to be highly inefficient ($\Delta p_{SUCCESS} <$ −0.8) almost 100% of agents exhibited compulsive episodes, supporting that a distrust in washing can be sufficient to induce compulsions. Interestingly, compulsive patterns could emerge even during phases when hands were objectively clean, thus appearing irrational to an external observer (see Fig 2B for an example sequence).

**Distrust in avoidance effectiveness is sufficient to cause compulsions.** Our simulations thus far included agents with various combinations of distorted beliefs. To isolate the impact of specific belief distortions, we implemented a subsequent set of simulations. In this set of simulations, agents had selective belief distortions such as a sole underestimation of the effectiveness of the avoidance behavior, while maintaining accurate beliefs about all other parameters of their environment (see Materials and methods –Simulation agents with selective belief distortions).

Fig 4B shows the results of this second set of simulations for a selective underestimation of washing success. The simulation confirms the link between washing success underestimation and symptom emergence and mimics the results of our first set of simulations (bootstrapped Spearman rank correlation in Table C in S1 Supplementary Material). Altogether, the underestimation of the avoidance action is thus sufficient to cause compulsive episodes, even in the absence of any other belief distortion.

**Overestimation of threat can trigger the onset of compulsive episodes.** We initially hypothesized that an overestimation of threat would also play a role in the development of compulsions. However, results from our full simulation did show no substantial or significant difference in the level of threat overestimation between compulsive and non-compulsive agents (bootstrapped $t$-test for threat magnitude ($cost_{SICKNESS}$) and probability ($p_{SOILING}$) in Tables 2 and A in S1 Supplementary Material). Furthermore, we found no significant correlation between the overestimation of threat and the severity of compulsions (bootstrapped Spearman rank correlation in Table B in S1 Supplementary Material).

In order to isolate the effects of threat overestimation, we conducted additional simulations with a selective distortion of threat magnitude and probability ($\Delta cost_{SICKNESS}$ and $\Delta p_{SOILING}$ respectively, see Materials and methods –Simulation agents with selective belief distortions). These targeted simulations revealed that while an overestimation of threat alone could initiate the onset of compulsive episodes (Fig 4C), the proportion and extent of those compulsive episodes remained low even with large belief distortions. Specifically, the compulsive episodes typically involved only 1–2 repetitive action repetitions, amounting to only a few percent of the total action sequence, Fig 4C). Notably, even with significant belief distortions, the overestimation of threat did not lead to prolonged or intense compulsive sequences comparable to those triggered by the underestimation of the effectiveness of the avoidance action.

Moreover, the overestimation of threat did not appear to influence belief updating processes, a point discussed further below (Fig 4C). Overall, it appears that while overestimation of threat can trigger the onset of compulsive episodes, it is neither necessary nor sufficient to sustain them in the absence of additional false beliefs.

## Predictions from the model

So far, our simulations identified an underestimation of the effectiveness of preventive actions as a key distinction between compulsive and non-compulsive agents. Next, we investigate how this belief distortion might predict and explain other characteristics of compulsive behavior.

**Underestimation of avoidance success leads to slowed belief updating and pathological doubt.** Building upon prior research indicating possible learning deficits in individuals with

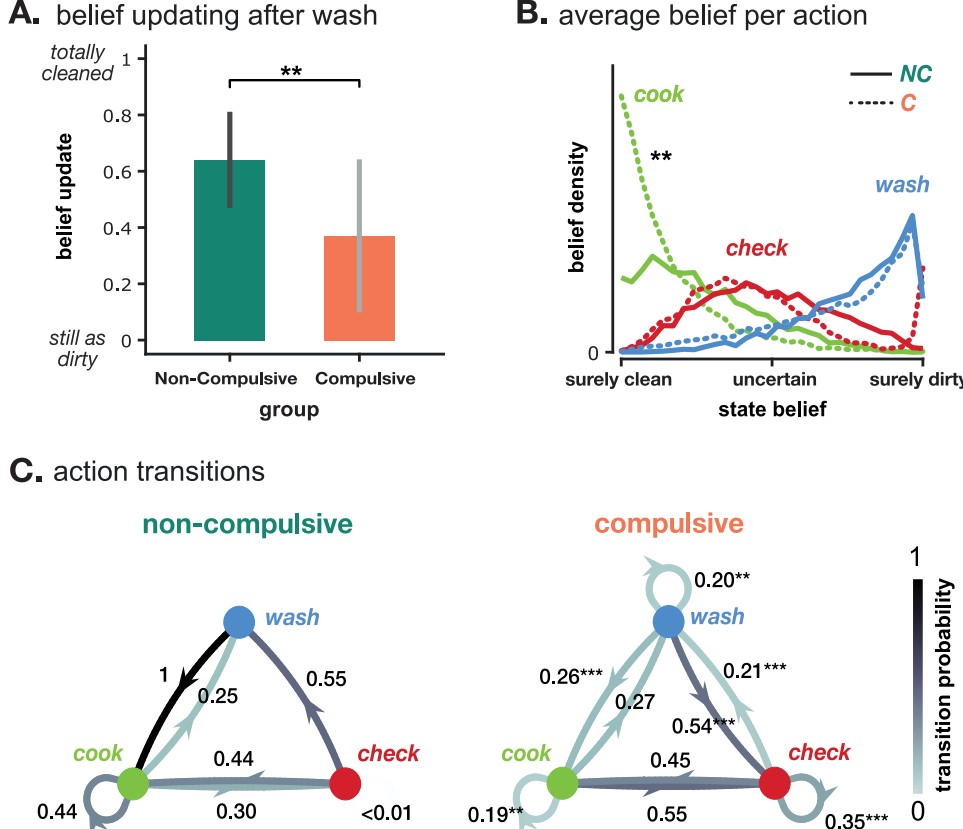

**Fig 5. Model predictions for belief updating, pathological doubt, intolerance to uncertainty and action repetition.**
**A.** Absolute size of the change in belief about the state of the hands (dirty or clean) after a wash action. Compulsive agents (C) have a significantly slower belief update than non-compulsive agents (NC). **B.** Distribution of the belief about the state of the hands when executing each of the three actions for the non-compulsive (NC, solid line) and compulsive (C, dashed line) groups. Large values indicate a large probability that an agent has the corresponding belief when executing the action. **C.** Action transitions for compulsive and non-compulsive agents. Frequency with which simulated agents transitioned (on average) from one action to another in the non-compulsive (left) and compulsive (right) groups. Arrows represent the empirical probability (indicated next to the arrow) of an action (end of the arrow) to be performed after another (start of the arrow), e.g. $p_{WASH \rightarrow CHECK}$, averaged across agents, states, and beliefs. Circular arrows represent action repetition probabilities. Darker colors represent higher transition probabilities. Notably, non-compulsive agents always return to cooking after washing, while compulsive agents are much more likely to wash or check again. * indicate significant group differences with p < *0.05, **0.01 or *** 0.001.

OCD (see e.g. [26,27]), we evaluated the belief updating mechanisms in compulsive and non-compulsive agents within our simulation. Specifically, we focused on how these agents updated their beliefs regarding hand cleanliness (Equation Eq 2). A key finding was that the compulsive group exhibited a markedly reduced amplitude in belief updates following hand washing compared to the non-compulsive group (non-compulsive group: belief update = 0.640 (0.171), compulsive group: belief update = 0.370 (0.272), bootstrapped *t*-test: *p* = 0.011; Fig 5A). This reduction indicates that compulsive agents experienced prolonged periods of uncertainty about the cleanliness of their hands and required extended sequences of checking or washing to shift their belief from 'hands are dirty' to 'hands are clean' (Fig 5A).

Interestingly, this decrease in belief update amplitude was more pronounced in agents with a greater underestimation of success probability (correlation with ($\Delta p_{SUCCESS}$: ρ = 0.728, *p* < 0.001). Put simply, the more an agent doubted the efficacy of their preventive washing

action, the slower they were to update their beliefs, leading to longer compulsive episodes. This lag in belief updating among compulsive agents suggests that compulsive episodes may represent a phase of heightened uncertainty or pathological doubt regarding the state of their hands, as depicted in (Fig 5B). It's crucial to underscore that this pathological doubt is not caused by uncertainty in the perception of the world, such as uncertain observations (captured by the parameters: $\Delta p_{DETECT,DIRTY}$, $\Delta p_{DETECT,CLEAN}$), but by an inherent distrust in the successful execution of actions ($\Delta p_{SUCCESS} < 0$). This kind of uncertainty corrupts about the state of the world and could potentially be a contributing factor to the learning deficits observed in individuals with compulsive tendencies.

**Underestimation of avoidance success leads to perfectionism/intolerance to uncertainty.**   OCD is often linked with heightened intolerance to uncertainty and tendencies towards perfectionism. Conceptually, this can be understood as a need for excessive certainty about the state of the world. For instance, while most individuals might be comfortable with a certain level of uncertainty (e.g., accepting a 80% probability of having clean hands), those with OCD-related perfectionism might require a much higher certainty level (e.g., being 99% sure their hands are clean) before engaging in subsequent actions, such as cooking.

To investigate this aspect, we analyzed the decision-making processes (action policies) of both compulsive and non-compulsive agents, focusing on their tolerance for uncertainty regarding hand cleanliness. We found that compulsive agents required a higher degree of certainty about the cleanliness of their hands before deciding to resume cooking compared to the non-compulsive group. This was evident in the average belief about hand cleanliness at the point of deciding to cook (Fig 5B, compulsive: b(clean) = 87.5% (12.3), non-compulsive b(clean) = 74.8% (17.4), bootstrapped U-test: $p = 0.010$). This behavior pattern indicates a marked level of perfectionism or a lower tolerance to uncertainty among compulsive agents (Fig 5B and Table D in S1 Supplementary Material).

**Compulsions are preventing exposure to true action outcomes.**   As compulsive behaviors are characterized by the repetition of certain action patterns, we next examined the probabilities of transitions between actions, such as from washing to cooking (Fig 5C). As expected from our definition of compulsions agents in the compulsive group were significantly more likely to repeat preventive actions like checking or washing (compulsive group: $p_{CHECK \rightarrow CHECK}$ = 0.35 (0.34), $p < 0.001$, $p_{WASH \rightarrow WASH}$ = 0.20 (0.318), $p = 0.002$; non-compulsive group: $p_{CHECK \rightarrow CHECK}$ = 0.004 (0.06), $p_{WASH \rightarrow WASH}$ = 0 (0.001), see bootstrapped U-tests in Table E in S1 Supplementary Material).

In addition, while non-compulsive agents typically resumed cooking immediately after washing their hands (NC: $p_{WASH \rightarrow COOK}$ = 1 (0)), compulsive agents often engaged in additional checking or washing actions before returning to cooking (C: $p_{WASH \rightarrow COOK}$ = 0.26 (0.42), $p < 0.001$). Notably, this behavioral reorganization was not observed for checking: both groups were equally likely to resume cooking after checking. This asymmetry is likely driven by the distrust in the washing actions. This implies that while controls experienced the reward of having cooked properly always right after washing their hands, compulsive agents were exposed to this reward only after having first checked—possibly multiple times—, thus stretching the link between washing and its positive outcome, which complicates credit assignment (see discussion below).

**Relative costs cause differences in compulsion types: Checking versus washing.**   Compulsions tend to cluster in different dimensions, such as checking, washing, ordering. In our framework we could analyze the individual trade-off between two of these behaviors, checking and washing. A common observation across all our simulations was that an underestimation of the effectiveness of preventive actions invariably led to compulsive episodes. While the degree of underestimation scaled with compulsion severity (in terms of probability, length,

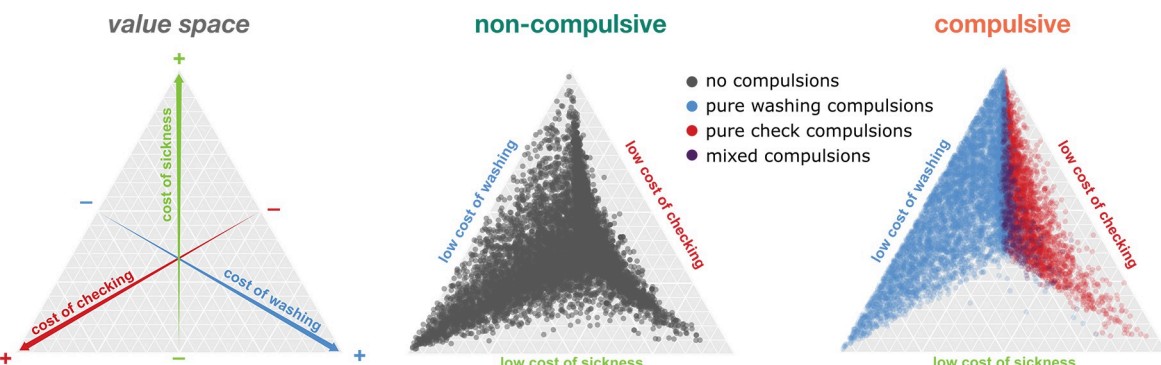

**Fig 6. Impact of subjective cost on compulsion types.** After normalization, the subjective costs of checking, washing, and cooking in the dirty state (cost of sickness) sum to one and can be represented as a dot on the canonical simplex (left). For example, an agent with an equal cost for the three actions would be represented exactly in the center; an agent with an extremely high cost of sickness relative to the cost of checking and washing would be located at the very top of the triangle representing the simplex. Non-compulsive agents (gray dots, middle) were spread almost all over the value space. The compulsive agents (colored dots, right) had a very similar distribution. Agents exhibiting pure washing compulsions (blue dots) all had a low cost of washing (left side of the simplex), while agents exhibiting pure checking compulsions (red dots) all had a relatively low cost of checking (right side). When the cost of washing and checking are balanced, agents were characterized by mixed compulsions (purple dots). Plotted agents were generated in the simulations 1A and 1B, here aggregated.

and frequency of episodes), the relative proportions of the different compulsion types (only checking, only washing, or a mix of both) remained consistent.

However, a comparison between different simulations that employed different criteria for compulsions (simulation 1A: requiring agents to perform all actions and simulation 1B: where checking was not mandatory; see Materials and methods Simulating compulsive and non-compulsive agents) revealed notable differences in compulsive behavior patterns. In simulation 1A a significant proportion of agents showed pure checking or mixed compulsions, with only 37.2% demonstrating pure washing compulsions. In contrast in simulation 1B the vast majority of agents (97.7%) exhibited pure washing compulsions.

To understand what explains the emergence of checking *vs.* washing compulsions, we analyzed the subjective costs attributed to these actions by agents in both simulations. We found a distinct pattern: agents exhibiting pure checking compulsions perceived a relatively lower cost of checking, while those with pure washing compulsions perceived washing as less costly. Agents who assigned similar costs to both actions tended to exhibit mixed compulsions (Fig 6 and Table F in S1 Supplementary Material). This finding indicates that the specific costs assigned to actions like checking, washing, or cleaning play a pivotal role in determining the nature of compulsive behavior, independent of the agent's confidence in their actions or the severity of symptoms.

## Discussion

In this study, we developed a computational model to explore the effects of dysfunctional beliefs on the emergence of compulsions. Our findings indicated that simulated compulsive agents differed from non-compulsive agents in one core belief only: They underestimated the effectiveness of their preventive action (here: handwashing to prevent illness). The extent of underestimation correlated with the prevalence and severity of compulsions. Moreover, the model sheds light on various phenomena related to OCD. It elucidates the role of other beliefs —such as doubts, perfectionism, intolerance to uncertainty and threat overestimation—in causing and maintaining compulsions. Furthermore, our findings offer insights into the learning deficits commonly observed in OCD patients and provide a deeper understanding of the

'not-just-right' experiences encountered by these individuals as discussed in further depth below (Fig 7).

## The role of dysfunctional beliefs for triggering and maintaining compulsions

Dysfunctional beliefs have been described as a core phenomenon of OCD in cognitive-behavioral theories and in the DSM [3,13]. The most common dysfunctional beliefs include pathological doubts, intolerance to uncertainty, and an overestimation of threat. Here we used a computational simulation to delineating the role of those beliefs for the emergence and maintenance of compulsions.

**Pathological doubt.**   Pathological doubt is a defining feature of OCD that earned it the nickname "la folie du doute" ("disorder of doubt"). Doubts are thought to be a major driver of persistent and unwanted intrusive thoughts (e.g. "What if my hands are contaminated?") and mistrust in one's own actions and observations (e.g. "Did I clean my hands properly?") [11,28].

To understand the role of pathological doubt in compulsions, it is crucial to distinguish between different types of doubt. In our simulations we separately assessed the role of doubts about actions (here, e.g. doubt about the success of handwashing), doubts about observations (here, the accuracy of observation during checking; see Fig 2 and Table 1), and doubts about the state of the world (here, the current belief about whether the hands are clean or dirty; see Fig 5B).

Our simulations suggest that compulsive behavior is only triggered and sustained when someone underestimates the effectiveness of their preventive action. Just merely doubting one's observations or perception was not enough to initiate a cycle of repetitive behavior. Furthermore, doubts about the state of the world directly emerged from the lack of faith in the effectiveness of the preventive action: If an individual distrusts the effectiveness of their preventive action, doubt persists after its execution and it can lead to a repetition of actions until the desired outcome feels to be achieved (e.g., ensuring the stove has been turned off properly, avoiding contamination, or locking the front door correctly). Our simulations showed that when agents were uncertain as to whether their handwashing was effective, it took multiple actions (washing and checking) to change their belief from thinking their hands were dirty to thinking they were clean (see Eq 2 and Fig 5B), whereby the exact pattern of washes and checks was determined by the individual costs of each action (Fig 6). This means that compulsive agents would linger longer in an in-between, uncertain, state where they were not convinced their hands were clean despite repetitive washing.

This in-between state may align with what patients frequently report as 'not-just-right' experiences or 'feelings of incompleteness' [29,30]. Individuals with OCD tend to be highly attuned to their repetitive behaviors and use the 'not-just-right' feelings as a prompt to perform the compulsion until they reach a sense of completion [30–33]. In our computational model this sense of completion would be achieved once the belief has transitioned from thinking the hands are dirty to believing they are clean enough to resume cooking. Thus, in our scenario, a lack of trust in the preventive action explained both why repetitive behaviors occurred and why they ended.

**Intolerance of uncertainty.**   While closely related to pathological doubts, intolerance of uncertainty emerges as a distinct, widely recognized symptom in OCD. Similar to doubts, intolerance of uncertainty can be defined in several ways [34,35]. In the most prevalent self-report measures of intolerance to uncertainty—the Intolerance of Uncertainty Scale [36] and the Intolerance of Uncertainty Scale Short Form (IUS-12) [37]—it is characterized as the propensity to react or think negatively toward uncertain events.

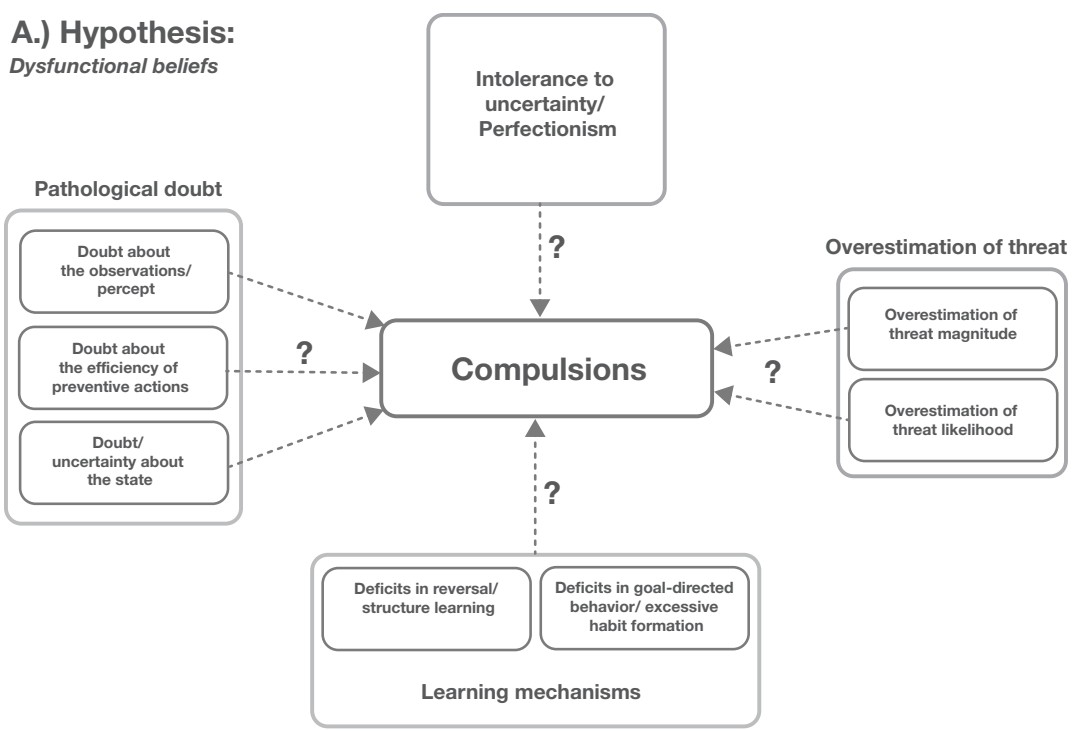

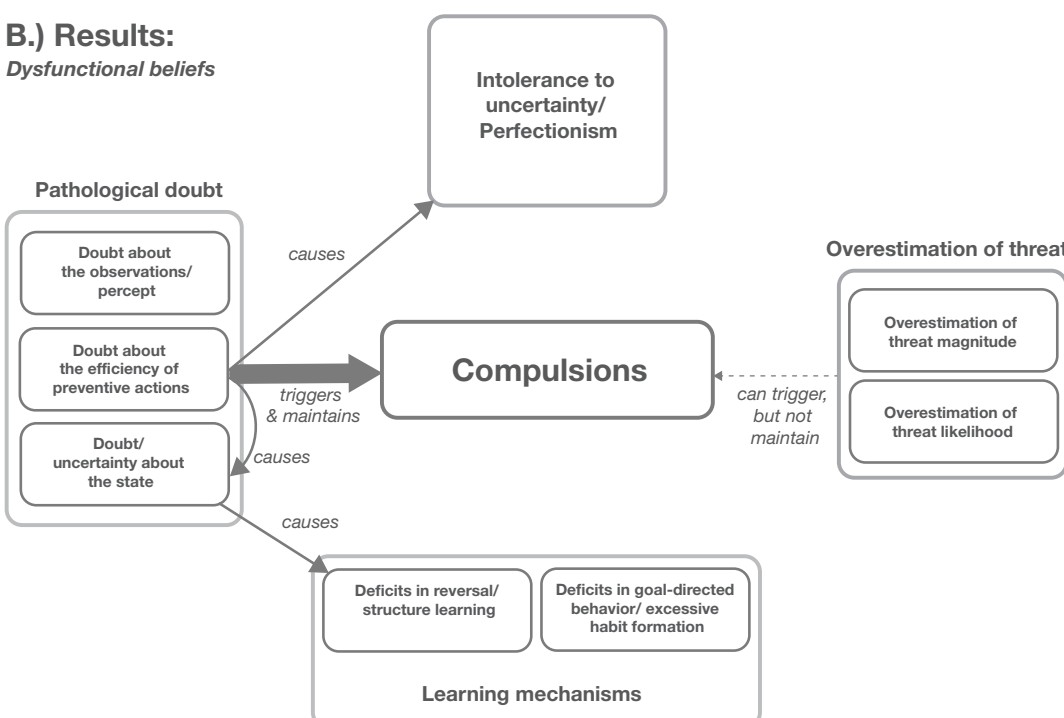

**Fig 7. Role of dysfunctional beliefs in compulsion formation. A.** Original hypotheses about beliefs that could lead to compulsions. **B.** Causes of compulsive behavior as suggested by our simulations. We found that in our simulations that compulsions were primarily triggered and maintained by doubts about actions, specifically a distrust in the effectiveness of an agent's active avoidance/ preventive behavior and not by any of the other dysfunctional beliefs (intolerance to uncertainty, overestimation of threat, doubt about observations etc.). An overestimation of threat could trigger the onset of a compulsive

episode, but did not maintain a compulsive episode. The distrust in the effectiveness of active avoidance in turn caused doubt and uncertainty about the underlying state of the environment which in turn can cause deficits in reversal and structure learning (see details in the Discussion).

Interestingly, while patients with General Anxiety Disorder and OCD generally show elevated levels on intolerance of uncertainty scales [12], research suggests a stronger association of these measures with anxiety rather than OCD [38]. This implies that while intolerance of uncertainty may provoke anxiety, it does not necessarily lead directly to compulsive behaviors.

Another perspective on intolerance of uncertainty aligns it more closely with perfectionism [39,40]. Here, it is seen as a drive to eliminate even the slightest uncertainty about an outcome. For instance, an individual with high intolerance of uncertainty or perfectionism might strive to increase the probability of a positive outcome from an already high 99% to an even more assuring 99.9%. This quest to minimize uncertainty could, in theory, be a driving force behind compulsions, compelling individuals to engage in repetitive checking until they attain a certain level of certainty. We investigated this theory in our simulations and found that simulated compulsive agents indeed adopted a stricter state certainty criterion in their policy: they waited to be more certain about the cleanliness of their hands to resume cooking (Fig 5).

**Overestimation of threat.** We further looked into the connection between an overestimation of threat and compulsions. Two types of overestimations of threat that appear to be especially pertinent for compulsions were identified and simulated: the size of the threat (threat magnitude), and the likelihood of the threat (threat probability) (see Table 1). For instance, if the perceived cost of a threat is very high (for example, having your car stolen because you did not lock it, or your house burnt down due to a lit candle), it logically follows that one might be more inclined to repeatedly check safety measures (assuring multiple times that the doors are locked or that the candle is extinguished). Similarly, a heightened perceived risk of contamination, especially in situations like a pandemic, could rationalize frequent hand washing. Hence, an overestimation of threat presents another plausible driver for compulsive behavior.

Our simulations specifically focused on scenarios where estimates of threat magnitude and probability were irrational, diverging from the actual risks present. We anticipated notable differences in threat beliefs between compulsive and non-compulsive agents. However, to our surprise, the simulations revealed no significant difference in threat overestimation between these two groups (although there was a trend, see Table 2).

Further, in a separate series of simulations where beliefs about threat magnitude or probability were exclusively manipulated, we observed that while an overestimation of threat could initiate checking or washing behaviors, these actions did not evolve into full-blown compulsions characterized by prolonged and repetitive behaviors. Crucially, this overestimation also had no discernible effect on belief updating, thus failing to account for pathological doubts or the extended state of uncertainty that might underpin lengthy compulsive episodes. This becomes clearer when we consider that having confidence in a preventive action, such as hand-washing, typically implies that performing the action once should be sufficient to mitigate the risk. It is primarily the presence of doubts about the effectiveness of such preventive actions that leads to their repetition. Therefore, while an overestimation of the threat may indeed exacerbate the symptoms by triggering the onset of compulsive episodes, it is the concurrent doubts about the effectiveness of preventive actions that are essential to sustain such recurring behaviors over time. In scenarios where these doubts are absent, even an exaggerated perception of threat alone is not enough to perpetuate compulsive actions (Fig 7B). As discussed below the role of threats and actions might play an important distinguished role in OCD and Generalized Anxiety Disorder.

Interestingly, in a study that asked OCD patients and healthy controls to report the incidence of OCD-related events, patients did not report higher incidence rates (threat probabilities) than healthy controls, yet patients did express a greater perceived vulnerability to those events [41,42]. This disparity might be attributed to their lower confidence in their ability to prevent the realization of such outcomes, despite having an accurate understanding of event probabilities. This finding underscores the complexity of threat perception in OCD, highlighting the need to distinguish between the actual likelihood of adverse events and the perceived ability to avert them.

## Link between theories of OCD

Several theories about the emergence of compulsions in both humans and animals have been proposed, each offering different perspectives on the origins of these behaviors. Here we discuss three major theoretical frameworks, examining how they may be linked to one another and at least be partially reconciled.

**Cognitive-behavioral theories.**   In the cognitive-behavioral theories of OCD, compulsions are seen as rational consequences of (potentially) irrational beliefs. An early proposal of a cognitive-behavioral theory of OCD posited that patients misinterpret the significance of normal, distressing and intrusive thoughts, which leads them to engage in compulsions, often to prevent harm for themselves or others [3,43]. For example, a patient might have a thought of harming someone by passing on germs that cause a fatal infection, which can result in excessive cleaning and checking of hands, avoidance of touch or interactions, excessive praying or another seemingly protective behavior. The theory highlights the role of an elevated sense of responsibility that drives a patient into acting [10,44]. Patients may, for instance, specifically worry about *not preventing* or *causing* harm, not about harm alone. Since the original proposal a large body of research assessed the role of additional dysfunctional beliefs in OCD [1]. Building on this work the Obsessive Compulsive Work Group (OCCWG) defined several core dysfunctional beliefs in OCD including an overestimation of threat and inflated responsibility, a need to control thoughts and perfectionism and intolerance to uncertainty [13,45].

The broad consensus around the cognitive theory of OCD is reflected in the definition of compulsions in the DSM-5 as *"repetitive behaviors (e.g., hand washing, ordering, checking) or mental acts (e.g., praying, counting, repeating words silently) that the individual feels driven to perform in response to an obsession or according to rules that must be applied rigidly. The behaviors or mental acts are aimed at preventing or reducing anxiety or distress, or preventing some dreaded event or situation. . .".* The DSM also specifically lists dysfunctional beliefs under the specification of the disorder [9].

One caveat of cognitive theories of OCD is that it is difficult to determine the specific impact of different dysfunctional beliefs on symptom manifestation. This is in part because the bulk of work relies on clinical and self-report instruments where patients typically report more than one belief distortion. Dissecting the contribution of different beliefs in an experimental setup in contrast is challenging.

Another way of dissecting the role of different dysfunctional beliefs for symptom manifestation is computational modeling [19,23]. Here, we used this approach to demonstrate that reduced confidence in the effectiveness of preventive (harm avoiding) actions can drive compulsions while an overestimation of threat and other forms of doubts and intolerance to uncertainty may arise as consequences of the former belief (see Discussion on The role of dysfunctional beliefs for triggering and maintaining compulsions, Fig 7).

**Habit learning theories.**   While in cognitive-behavioral theories of OCD compulsive actions are typically thought to fulfil a concrete goal (e.g. avoiding harm), an alternative

influential theory proposes that compulsions are not goal-directed behaviors but the results of maladaptive habit learning [5,46].

When actions are performed with some regularity the relationship between the stimulus and the response is strengthened which can lead to the formation of habits [47]. Habits are automated behaviors that form solely on the grounds of previous experiences. They do not require a specific action goal or plan. For example, when operating a manual vehicle, shifting gears starts out as a goal-directed action, however, with time it becomes more habitual.

The habit theory of OCD postulates that compulsions result from a disruption in the balance between goal-directed and habit learning, which could result either from excessive habit formation (strong stimulus-response associations) or a failure of goal-directed control (to overwrite a habitual response) [5,48,49]. Obsessions are then thought to be post-hoc rationalizations of habitual behaviors and not the drivers behind compulsive behavior [5,50].

This habit theory of OCD was partially inspired by neurobiological findings in rodents and humans that indicated an involvement of fronto-striatal circuits in both OCD and the transition from goal-directed behavior to the formation of habits [51,52]. Habit formation in OCD was predominantly tested using devaluation [48,53] and reversal learning paradigms [27,54,55] where perseverative responding after a contingency change was taken as a sign of habit formation.

The evidence for these perseverance errors in OCD is mixed. While there are signs of continued responding after devaluation ([48,53], but see [56]), errors in reversal learning appear to be more spontaneous [55,57]. Furthermore, in the two-step task, a paradigm specifically designed to assess the balance between habitual and goal directed action selection [58], patients with OCD patients displayed no differences in habitual learning for loss outcomes and effects on reward learning seemed to be primarily driven by chronic SSRI use [59]. In a large online study using the same task, OCD diagnosis was not associated with decreased goal-directed performance, instead the authors found a correlation with transdiagnostic self-reported compulsivity [60]. These empirical results point to specific learning deficits in OCD but leave the causal role of habit formation unresolved.

**Linking dysfunctional beliefs, habits, and learning.** While the cognitive-behavioral theory provides a comprehensive explanation for many common clinical observations in OCD, it falls short of offering a systematic framework to account for several frequently observed experimental phenomena. This includes the lack of a clear explanation of how dysfunctional beliefs lead to perseverance in reversal learning and devaluation paradigms.

In comparison, the habit learning theory accounts for the perseverance errors in patients by referring to their reliance on habitual responses instead of goal-oriented control. However, empirical tests of habit formation have yielded mixed results (see Discussion Habit learning theories). Moreover, this theory cannot fully explain other aspects of the disorder [61], such as why compulsions typically focus on avoiding unfavorable outcomes instead of also occurring in rewarding contexts, and why habits seem to be at odds with patients' close monitoring of their actions, as evidenced by the presence of not-just-right-feelings, a heightened sense of responsibility, and heightened error-responses. Additionally, the notion of obsessions as post-hoc rationalizations does not explain why responsibility manipulations can trigger urges to engage in checking [62,63].

In a recent theoretical paper, Fradkin et al. (2020a) challenged the notion that habits alone are responsible for the errors made by OCD patients during reversal learning and devaluation tasks [6]. Instead, their computational framework (active inference POMDP) is based on the idea that difficulties in detecting and predicting changes in the environment are the underlying cause for such failures [64]. The framework suggests that excessive uncertainty about state transitions (e.g. from clean to dirty hands or from one task contingency to another) can

explain various observations in OCD, including the formation of excessive checking and learning deficits [6]. The authors propose that state uncertainty might cause patients with OCD to resort to simpler, less computationally intensive strategies, resulting in habitual, perseverative behavior when the environment is stable (e.g. after over-training in a devaluation task). Conversely, in less stable environments (e.g. during reversal learning paradigms), it causes patients to adopt more exploratory strategies. This balance based on state uncertainty can explain seemingly conflicting findings that suggest patients with OCD persevere in some scenarios [48] and exhibit exploratory behavior in others [54,65] and has also been linked to anxiety disorders in general [66].

Our model offers a nuanced expansion of the concept of state uncertainty, particularly in its relevance to OCD. We find that the form of uncertainty pivotal to OCD is specifically linked to state-action transitions concerning harm prevention. This focus is distinct from a broader, generalized uncertainty about state transitions or other kinds of action-state interactions, such as those related to incidental contamination (e.g. likelihood of contaminating hands while cooking).

Furthermore, our simulations show that compulsions in OCD are not propelled by a uniformly elevated level of uncertainty around these action transitions. Instead, they stem from a precise, asymmetrical underestimation of the success in harm avoidance activities. This further suggests a context-specificity of compulsions for OCD that are specific to preventive actions for harm avoidance.

Crucially, we show that this unique underestimation of successful harm prevention can serve as the root cause for other forms of uncertainty and doubts, including those related to causing uncertainty in state transitions as they have been proposed in Fradkin et al. 2020 [26] (see Figs 5B and 7). In essence, it is this specific and asymmetrical distrust in harm prevention that potentially underlies and explains the broader spectrum of uncertainties, doubts, and dysfunctional beliefs observed in OCD (Fig 7). As a result, our model generates several concrete and testable predictions, which are outlined below.

## Ideas and speculations

**Testing some the model predictions.**   Our model posits several predictions that could be empirically tested.

First, it predicts that patients should show slowed learning and detection of state transitions (as also proposed by [6]). Evidence for slowed transitions between states has been observed in devaluation tasks [48,53] but also in paradigms that more explicitly tested state learning such as [26,67,68]. In the latter the authors found that belief stickiness and a decreases in the propensity to learn about a state transitions, in their case a change of virtual seasons, correlated directly with the degree of obsessive-compulsive traits and was improved by a single dose of an SSRI, the most common drug treatment for OCD [68].

Alongside the previously observed slowed learning and detection of state transitions in, our model uniquely predicts differences in tasks involving active action selection compared to passive learning (e.g. [69]). Notably, an EEG study found that OCD patients exhibited reduced N1 suppression to actively generated feedback compared to passively observed feedback, correlating with enhanced feelings of agency and incompleteness [70]. Similarly, changes in self-agency were noted in OCD patients using a "gaze-agency" task [71].

Another critical prediction of our framework is the goal-oriented nature of avoidance actions in OCD, wherein the precise execution of these actions is vital. This heightened monitoring could account for the altered cognitive processes in OCD, such as the enhanced error-related negativity (ERN). ERN, a neurophysiological marker for the incorrect execution of

actions, has been consistently found to be elevated in OCD patients [72–75] and their first-degree relatives [76], with a direct correlation to symptom severity [72].

Additionally, the model predicts that increasing the threat level should elevate the frequency but not the length of compulsive episodes. This implies that while manipulating perceived threat levels may alter the patterns of compulsive behavior, it may not address the underlying cognitive distortions.

To directly assess beliefs regarding successful harm prevention, one could develop a dedicated task based on the simple handwashing scenario outlined in this study or a similar task design [77]. To ensure empirical feasibility and computational tractability, it would likely be necessary to select a specific instance of world contingencies (world parameters) that optimize parameter recovery for the limited number of trials that can be acquired. While this paradigm should effectively uncover beliefs associated with successful harm prevention ($p_{SUCCESS}$), thorough simulations would be required to evaluate parameter recovery across the various other parameters.

In addition to implementing the proposed paradigm, one could also introduce a variant where avoidance actions are either actively chosen by participants or selected by a different agent. Comparing policies between these two paradigms would provide valuable insights, as discussed earlier.

**OCD as a disorder of control and responsibility.** An intriguing aspect of our findings is the apparent link between beliefs about controllability–defined as the likelihood that an action will lead to a desired outcome–and the occurrence of compulsive behaviors. Our simulations reveal that a perceived lack of control, specifically underestimating one's ability to successfully prevent harm, can be a potent driver of compulsive behaviors. This aligns with the notion that compulsions are, in part, efforts to regain a sense of control over perceived threats or negative outcomes.

The findings prompt an important question: What underlies this compelling need to exert control over negative outcomes? A likely explanation is a heightened sense of personal responsibility for causing potential harm. This intensified sense of responsibility is a characteristic feature in OCD, known to fuel compulsive behaviors [10,44,78]. Studies involving manipulations of perceived responsibility have shown that an increased sense of responsibility amplifies discomfort and the urge to check in OCD patients, whereas lower perceived responsibility yields the opposite effect [63].

This heightened sense of responsibility could also explain both rational and seemingly irrational compulsive behaviors. While washing to prevent harm seems to be a rational preventive behavior, patients also often engage in irrational, ritualistic actions like tapping, driven by spurious beliefs that these actions can prevent negative outcomes. Such idiosyncratic associations between actions and outcomes could arise from an ingrained belief in personal responsibility, fostering a strong impulse to perform actions believed to counteract potential harm, regardless of their actual efficacy or logical connection to the outcome. The awareness of the implausibility of these links could further heighten the distrust in the effectiveness of these preventive actions.

Our model implicitly incorporates a notion of responsibility by assuming a link between preventive actions and their believed effectiveness in averting harm and could thus in principle also account for spurious links between actions and outcomes. Yet, it stops short of explaining the origin of the pathologically inflated sense of responsibility in OCD. One speculative thought could be that believing that one is unable to prevent bad outcomes makes individuals more likely to believe that they will cause harm, which results in a sense of responsibility. How responsibility is linked to beliefs about harm prevention remains an open area for future investigation, offering potential insights into the complex cognitive mechanisms underpinning OCD.

                                         

**OCD and anxiety disorders.** The role of control and responsibility may be what differentiates OCD from other anxiety disorders (ADs). Both OCD and ADs are characterized by intrusive thoughts and worries and behavioral strategies aimed at avoiding distressing outcomes. Yet, the perception and execution of these strategies differ between the disorders. In OCD, perceived threats ignite a sense of responsibility and an urge to act preventively, leading to compulsions that often fail to align rationally with the perceived danger (active avoidance). In contrast, other ADs typically involve avoidance behaviors aimed at circumventing stressful situations, which may correctly anticipate a loss of control. A similar computational approach to the one used in this paper has recently been used to capture this type of avoidance in anxiety disorders [79]. In the context of our model, this aligns with cases of selective overestimation of threats, which lead to an increase in avoidance actions but not to sustained compulsions.

## Model assumptions & limitations

**Assumptions.** One core assumption of our model is that agents act rationally (choosing an optimal policy) under their own—potentially false—beliefs. Importantly, the ensuing behavior can nevertheless appear irrational to an outside observer who holds a different set of beliefs. For instance, in contexts where health threats by contaminated hands are regarded as unlikely it may be considered irrational to excessively and repeatedly wash one's hands. However, excessive handwashing may be regarded as a rational behavior during a pandemic with a highly infectious novel virus. In our simulation this assumption, that agents act rationally under their own beliefs, is rooted in the particular choice of the model: A POMDP which predicts the optimal policy under the given contingencies of the world. Our specific tweak to that model was that we allowed individuals to have their own assumed contingencies of the world which can deviate from the structure of the real world from which they derived their policy. Importantly, no matter what the internal belief was, all agents were interacting with the real world. In that way we could assess how an individual's beliefs can trigger compulsions in an interaction with a real world where their beliefs were invalid.

**Limitations.** Our computational approach has several limitations. Firstly, we examined one particular manifestation of compulsions: repeated washing and checking patterns. Secondly, the chosen scenario of safety behavior was deliberately kept simple. While it allows for constructing a minimal model and a full exploration of the parameter space, it does not guarantee generalization to all other clinically relevant scenarios. For the sake of simplicity, we also focus on the direct effect of parameters on the simulated symptoms instead of potential nonlinear influences of higher-level interactions. Third, we focused on compulsive behavior, not obsessions, here. Fourth, we assume one particular class of model (POMDP) that describes optimal decision-making under uncertainty.

Recent studies suggest that OCD patients fail to update their confidence about their actions [80]. To delineate the effects of confidence from bias we would include uncertainty around the belief estimates into our model. To do so one could consider a Bayesian formulation. Such an approach could in principle naturally extend the framework presented here, but would pose technical challenges that would render large scale simulations intractable.

Finally, our model does not describe how those individual beliefs about contingencies in the world (agent parameters) are acquired. This means it cannot discern between different developmental aspects of OCD, e.g. explain differences in onset and comorbidity between pediatric and adult OCD [81]. While a Bayesian reformulation of our model could potentially help capture such dynamics, we can at this point only speculate that during childhood and early adolescence a model of the contingencies of the environment is forming that becomes more rigid with age.

### Conclusion and outlook

The computational approach described in this paper suggests a specific cognitive mechanism of compulsive behavior: an underestimation of the probability that one's preventive actions are successful (action effectiveness). We find that this belief can trigger and maintain compulsive episodes and scales with symptom severity. Further, it results in several downstream effects, such as increases in other forms of pathological doubt, intolerance to uncertainty and state learning deficits. If these *in silico* results can be confirmed by empirical studies with suitably designed cognitive tasks and generative models, this may open new avenues for stratifying OCD patients into mechanistically interpretable subgroups and support individualized deployment of cognitive therapy with an increased focus on treating beliefs about agency and control in OCD.

## Materials and methods

### Computational model

We used a normative model of decision-making under uncertainty, a Partially Observable Markov Decision Process (POMDP), to describe the handwashing scenario (Fig 2). POMDPs represent a good approximation to many real-world scenarios by describing how agents would act optimally, in relation to their individual beliefs, in the presence of uncertainty [82].

**Simple safety scenario.**   We chose a simple safety scenario to assess the impact of beliefs on compulsions that is typical for OCD. In this handwashing scenario, we let simulated agents choose between three actions (cooking, washing, checking) at any point in time while either having clean or dirty (contaminated) hands (Fig 2A). If they decided to cook, there were two possible outcomes: a successful dinner (positive outcome) if the hands were clean or sickness of their guests (detrimental outcome) in case the hands are dirty. The state of their hands could change over time depending on the chosen action: They could get dirty while cooking (with probability $p_{SOILING}$) or become clean again when they chose to *wash* their hands (with probability $p_{SUCCESS}$). Critically, we assumed that our simulated agents could not directly observe the state of their hands. However, they could *check* their hands to gain additional, yet indecisive, information about whether the hands were dirty or clean and thus improve their predictions about the outcomes. Ideally, they thus carefully weighed the risk of *cooking* with dirty hands against the cost of *checking* and/or *washing* to stay clean to host many successful dinners while avoiding sickness. The optimal decision sequence was thereby the one that maximized the long-term total net outcome over many dinners.

In our specific formalization of the POMDP model, we assumed that every agent not only interacted with an environment with fixed rules (fixed world parameters), but also based on their individual beliefs about the world (their agent parameters). This allowed us to simulate agents that make optimal choices under potentially false beliefs. The code for the model can be found here: *https://github.com/lionel-rigoux/beliefs-compulsions-and-reduced-confidence-in-control*.

**Markov decision process.**   Markov Decision Processes (MDP) provide a formal framework for modeling sequential decision making as a value optimization problem. Formally, we first define a discrete set of states $s \in S$ that represent potential realizations of the world, i.e., external constraints imposed on an agent at a given time. At each time step, the agent must choose an action $a \in A$ that will (1) change the state of the world according to a transition probability $T(s'|a,s)$, and (2) yield a deterministic outcome as prescribed by a so-called "cost" or "reward" function $R(a,s)$ that also depends on the state. The core problem of MDP is to find

**Table 3. Transition probabilities T(s'|a,s) for different actions.**

| Action (a) | State (s) | Next State (s'): clean | Next State (s'): dirty |
|---|---|---|---|
| cook | clean | $1-p_{SOILING}$ | $p_{SOILING}$ |
| | dirty | 0 | 1 |
| wash | clean | 1 | 0 |
| | dirty | $p_{SUCCESS}$ | $1-p_{SUCCESS}$ |
| check | clean | 1 | 0 |
| | dirty | 0 | 1 |

the action policy $\pi^\star(s)$ that maximizes the long-term discounted cumulative sum of reward:

$$\pi^*(s) = \arg \max_{\pi(s)} \sum_{t=0}^{\infty} \gamma^t R(\pi(s_t), s_t) \tag{1}$$

where $\gamma \in [0; 1]$ is a discount factor that defines the time horizon of the policy. It captures the temporal discounting of future rewards: a lower value indicates a shorter planning horizon, t. e., being less influenced by the future consequences of actions. The main difficulty when solving equation (Eq 1) is to marginalize the reward function across all possible state trajectories that the policy could induce given the stochastic state transition $T$ (see Table 2). In our case this was computed numerically using dynamical programming methods [83,84] (see below).

**Specifications for the handwashing scenario.** In the implementation of our simple safety behavior scenario (Fig 2), there are two states: clean hands or dirty hands (state space: $S$ = {clean, dirty}), and three possible actions: cooking, washing and checking (action space: $A$ = {cook, check, wash}). The wash and cook action can lead to a transition from one state $s$ to the next state $s'$ as indicated by the transition probability T(s'|a,s) (see Table 3 for an overview of the possible transition probabilities).

As can be seen from Table 3 the *check* action never changes the state. Each action is associated with pay-offs, as captured by a reward function $R$. Importantly, we consider here that this reward function reflects the true utility of actions which might differ from the one perceived by the agent due to its idiosyncratic sensitivity to costs and benefits (but see below *Modeling action sequences for individual agents*). In our example, the reward function assigns a fixed (negative) outcome to the *wash* action, $R(wash,s) = cost_{WASH}$, and the *check* action, $R(check,s) = cost_{CHECK}$, which is independent of the state $s$. For the *cook* action, the outcome depends on the state. It is rewarding (positive) if the hands are clean, $R(cook,clean) = reward_{DINNER}$, and punishing (negative) if they are dirty, $R(cook,dirty) = cost_{SICKNESS}$. See Table 4 for an overview.

As the solution of the MDP is invariant under a linear transformation of the reward function, outcomes are evaluated only relative to each other. Therefore, this set of free parameters overspecifies the reward function. To address this issue, we fixed the only positive outcome,

**Table 4. Reward function R(a, s) for different actions and states.**

| Action (a) | State (s) | Outcome |
|---|---|---|
| cook | clean | $reward_{DINNER}$ |
| | dirty | $cost_{SICKNESS}$ |
| wash | clean | $cost_{WASH}$ |
| | dirty | $cost_{WASH}$ |
| check | clean | $cost_{CHECK}$ |
| | dirty | $cost_{CHECK}$ |

$reward_{DINNER}$, to an arbitrary value (here, 0) which serves as a baseline against which all other costs are compared, and normalized the costs such that they summed to unity ($cost_{CHECK}+cost_{wash}+cost_{sickness} = -1$). Note that this normalization does not constrain the parameter space.

**Partial observability and state uncertainty.** The solution of the MDP problem described in the previous section is trivial if the state is known: One should cook if the hands are clean, and wash them otherwise. However, we will now assume that agents cannot observe the true state of the world directly. Instead, they maintain a belief about the possible states, and decide upon the best action based on this belief rather than based on a known objective state. State beliefs can be updated either by anticipating the consequences of one's actions in a given state (for example getting dirty when cooking, getting clean when washing), or by actively probing the state of the world using the third action, *checking*. This action provides additional, yet incomplete, information about the state of the hands. For example, they *appear* clean or dirty. Now, agents can use this information to update their beliefs about the true underlying state of their hands. Formally, this problem corresponds to a so-called Partially Observable Markov Decision Process (POMDP) which is a generalization of the MDP framework [85].

The first extension is to define a belief space $B$ that supports the belief $b$ about the identity of the current state (clean or dirty). In our case, a belief is simply defined as the (subjective) probability, e.g. that the hands are dirty, $b = p(s = dirty)$. The set of all possible beliefs that define the belief space is, therefore, the line segment (or 1-simplex) $B = [0; 1]$.

This belief about the state should not be confused with subjective beliefs about the world parameters (agent parameters). While the latter are free parameters that we fix to define individual agents (Table 1), the state beliefs unfold in time at each action. Agent parameters are thus static point estimates. The belief about the state of the hands is a time-dependent distribution, albeit represented by a single (time-dependent) scalar value.

Second, the true state of the environment can influence the internal belief through observations $o \in \Omega$. Observations are elicited by the actions as a function of the state, as defined by the observation function $O(a,s)$. In our model, we included two observations, $\Omega = \{dirty, clean\}$ which are only informative for the check action (Table 5); for the wash and cook actions, observations are evoked at chance (0.5) and are therefore uninformative, *i.e.* while defined, they effectively do not affect the belief update (see below Eq 2).

The critical constituent of our POMDP model is the belief update rule that describes how beliefs, predictions, and new observations are combined to form a new belief after an action has been performed. We used the canonical Bayes' rule which prescribes how beliefs are updated in a statistically optimal manner:

$$b_{t+1} = p(s_{t+1} =' dirty' | a_t, o_t, b_t) = \frac{1}{Z} O(o_t | a_t, s_{t+1} =' dirty') \sum_{s \in S} T(s_{t+1} =' dirty' | a_t, s_t) b_t \quad (2)$$

where $Z = \sum_{s_{t+1} \in S} O(o_t | a_t, s_{t+1}) \sum_{s \in S} T(s_{t+1} | a_t, s_t) p(s_t)$ is a normalization factor.

**Table 5. Observation probabilities O(o|a,s) for different actions.** Abbreviation: Obs. Prob.is Observation Probability.

| Action (a) | State (s) | Obs. Prob. (o = clean) | Obs. Prob. (o = dirty) |
|---|---|---|---|
| cook | clean | 0.5 | 0.5 |
| | dirty | 0.5 | 0.5 |
| wash | clean | 0.5 | 0.5 |
| | dirty | 0.5 | 0.5 |
| check | clean | $p_{DETECT\ CLEAN}$ | $1-p_{DETECT\ CLEAN}$ |
| | dirty | $1-p_{DETECT\ DIRTY}$ | $p_{DETECT\ DIRTY}$ |

An intuitive interpretation of this equation is that the current belief ($b_t$) is used as a starting point to make predictions about the next state given the expected effect of the chosen action $a_t$ (summed over possible transition probabilities, $T$). This prediction about the next state is then updated by taking into account the relative probability of indeed making the observation ($o_t$) in that new state. Note that, in our model this means in case of a *wash* or *cook* action, the belief is only updated on the basis of internal predictions since for these actions the observation is uninformative (see Table 5). In contrast, the *check* action induces an update only based on the observation, as the state is not expected to change for this action.

**Policy optimization.**    As in the POMDP framework the true state of the world or environment is not accessible, an optimal agent must base its policy entirely on its belief. Formally, this amounts to rewriting the definition of the optimal policy defined by equation (Eq 1) by re-indexing the policy on beliefs rather than states, and marginalizing the reward function over possible states:

$$\pi^*(b) = \arg \max_{\pi(b)} \sum_{t=0}^{\infty} \gamma^t E[R(\pi(b_t), s_t) | b_t] \qquad (3)$$

where $E$ denotes the expectation over states given the current belief. Of note, although not explicitly conditioned on observations, the optimal policy still indirectly depends on past observations through their cumulative influence on the belief about the state.

Although this new formulation is intuitively very similar to the MDP optimal policy, its solution is substantially more difficult to compute. Indeed, while the state space is discrete, the belief space is continuous, and so is the solution. To find the optimal policy for our model, we used the *pomdp-solve* program written in C (v5.4, *http://www.pomdp.org/code/index.html*) developed by Anthony Cassandra.

This toolbox approximates the optimal policy using dynamical programming by iteratively adjusting a set of linear action-value mappings [82]. Solutions of the *pomdp-solve* software were then parsed using homemade scripts to perform follow-up analyses in Matlab (Version 9.3.0.713579 (R2017b). Natick, Massachusetts: The MathWorks Inc.). The code can be found on GitHub (*https://github.com/lionel-rigoux/beliefs-compulsions-and-reduced-confidence-in-control*).

**Modeling action sequences for individual agents.**    The optimal solution described by the POMDP depends on the parameters of the task: the probabilities of state transitions, the size of the outcomes, the observation probabilities for each action and the discount factor (see Table 1 for a list of the free parameters). Critically, we assume that agents do not know the parameters of the environment. Instead, they act based on their own subjective representation about the world parameters (agent parameters), which may differ substantially from the true parameters.

In order to simulate the effects of misrepresentations of the environment on behavior, we duplicated all parameters defining the POMDP problem. One set represented the subjective beliefs about the world parameters (agent parameters) and was used to compute the optimal policy and the belief update rule. The other set described the true world dynamics (world parameters) and was used to sample actual state transitions and observations. To avoid confusion, we denote subjective parameters with prime (′) (Table 1). For example, an agent who is overly sensitive to threats, and thus overestimates the expected cost of cooking with dirty hands, will be modeled with an agent parameter $cost_{SICKNESS}'$ that is larger than the objective cost (world parameter) $cost_{SICKNESS}$.

The pseudo algorithm for generating action sequences was:

- Initialization:

- find optimal policy $\pi^{\star\prime}$ for $\{T',R',O',\gamma'\}$

- draw random initial state and belief, $s_0$ and $b_0$

- For t = 0 to N

- select subjectively best action $a_t = \pi^{\star\prime}(b_t)$

- draw next state $s_{t+1}$, reward $r_t$, and observation $o_t$ according to the respective objective densities $T(a_t,s_t)$, $R(a_t,s_t)$, and $O(a_t,s_{t+1})$.

- update internal belief using equation (Eq 2) in which the objective transition and observation functions $T$ and $O$ are replaced by the subjective mappings $T'$ and $O'$.

Here, t stands for trial and indicates one instance of a decision event (cook, wash, or check). This combination of data from the real world and subjective beliefs in the POMDP allows us to predict the behavior of different agents with different belief sets, all of whom interacted with the same environments.

## Simulations

**Random parameter generation.**   We sampled the parameters of our POMDP by allowing them to take any possible (valid) value, ensuring a complete exploration of the parameter space in our simulations. Transition probabilities were uniformly sampled over their entire support, i.e. [0, 1]. For observations, as state inference is symmetrical around chance level (p = 0.5), probability of the observation being correct was only sampled over [0.5, 1]. For action values, defined by the set $\{cost_{WASH}, cost_{CHECK}, cost_{SICKNESS}, reward_{DINNER}\}$, we used the fact that a POMDP policy is invariant to linear transformations of the value function. Using the linear mapping

$$z(v) = \frac{v - reward_{DINNER}}{cost_{WASH} + cost_{CHECK} + cost_{SICKNESS}} \tag{4}$$

we have:
$z(reward_{DINNER}) = 0$
$z(cost_{WASH}) \in [0, 1]$
$z(cost_{CHECK}) \in [0, 1]$
$z(cost_{SICKNESS}) \in [0, 1]$
$z(cost_{WASH}) + z(cost_{CHECK}) + z(cost_{SICKNESS}) = 1$
In other words, $[Z(cost_{WASH}),Z(cost_{CHECK}),Z(cost_{SICKNESS})]$ is a point on the unit 2-simplex. By sampling uniformly over the 2-simplex, we therefore exhaust all possible parametrizations of the value function.

**Simulating different worlds.**   In the previous section we mentioned that the sequence of decisions depended both on the subjectively optimal policy (the solution of the POMDP according to the internal representation of the world by the agent) and the true contingencies of the world (the actual state transitions and observations).

In the first simulation we aimed to select a large variety of different worlds (representative parameter combinations of the nine world parameters) (see Fig 1A). In principle, it is now conceivable to create worlds where excessive washing is the rational and optimal thing. For example, this paper was finalized over the course of a global pandemic where a highly infectious virus made excessive handwashing a perfectly rational behavior. Yet for the purpose of this paper, we were interested to see when excessive handwashing and checking arise as a pathological consequence of *false* beliefs about the world, that is, in worlds where contamination is

unlikely and most individuals would not act in a compulsive fashion. Thus, we exclusively selected worlds (defined by the nine *world parameters*) in our model formalization in which repeated checking and washing was not optimal.

Thus, when we simulated parameter combinations of the world (world parameters) we systematically excluded environments in which 'compulsions' (successive repetitions of *wash* or *check* actions) were the optimal behavior. We did that by repeatedly sampling random sets of parameters for which we then simulated 200 sequences of 1,000 actions according to the corresponding optimal policy of the POMDP (see details in *Materials and methods—Random parameter generation*). We then selected 100 parameter combinations (worlds) where the *optimal* policy a) included all three actions (check, wash, cook) and 2) did not contain any compulsive episodes (repeated washing and checking).

**Simulating compulsive and non-compulsive agents.** Next, we simulated the decisions of many different agents that interacted with these environments under entirely different beliefs. Critically, we assumed that the agents did not know the real contingencies of the environment, but instead had their own subjective beliefs about the world parameters (*agent parameters*); notably, these beliefs could deviate substantially on an agent-by-agent basis from the parameters of the actual environment (*world parameters*) while the overall range of beliefs was not outside the range of true world parameters. Note that the subjective belief about each parameter is a point estimate here, not a distribution.

Our goal was to assess the link between belief distortions and compulsive behavior. In order to stratify our agents into two groups–*compulsive* and *non-compulsive*–we then computed, for each agent, the (subjectively) optimal policy, and simulated 200 sequences of 1,000 actions by letting the agent interact with a given world, and verified if the agent showed any compulsive episode.

A compulsive episode was thereby defined as any sequence of non-cook actions that included at least one repetition of either a wash or check actions. So, a *cook-check-wash-check-cook*, *cook-wash-wash-cook*, *cook-check-check-cook* would be considered compulsive episodes, because a non-cook action was repeated more than once. On the other hand, *cook-wash-check-cook* would not be considered a compulsive episode, because none of the actions besides cooking was repeated more than once (see Fig 2B for an example of an action sequence of a compulsive and non-compulsive agent). Any agent displaying at least one episode of a compulsive action sequence was assigned to the *compulsive* group. We continued the simulation until we had collected 50 compulsive and 50 non-compulsive agents for each of the 100 worlds, resulting in a total of 10,000 simulated agents.

Note that in reality, there is no reference or cut-off which number of repetitions would be considered to reflect compulsive versus normal behavior. In fact, the definition of a compulsion in OCD is highly context dependent and does not only depend on the number of repetitions of a behavior, but also its duration, frequency and associated psychological burden. Here, we thus deliberately chose an inclusive criterion for compulsions, as we were interested in belief distortions which contributed to any form of repetitive behavior. In a second step, we analyzed the degree to which a belief distortion related to length and number of compulsive episodes (see analysis on the relationship with symptom severity). This provided us with an indication which beliefs contributed to more severe forms of compulsive behavior (as would be selected by a more stringent definition of compulsions).

Despite the length of a compulsive episode, one could also argue that the type of compulsion might make a difference. In other words, if individuals only wash, or check and wash. To address this, we show the results for simulations with two different constraints on the action policy:

**Simulation 1A (full, all actions required):** All beliefs were varied, but only agents including all three actions (wash, check, cook) in their optimal policy were included. That is, an agent had to check at least once in their entire action sequence. This collection of 10,000 agents is referred to as the "mandatory checking" dataset.

**Simulation 1B (full, checking not required):** All beliefs were varied, in this case, we also included agents whose policy did not include the check action, so agents that never checked at all. This collection of 10,000 agents is referred to as the "relaxed checking constraint" dataset.

Note that the two datasets were simulated on the same worlds. Since the results are extremely similar, we focused on Simulation 1A for the main figures but also show Simulation 1B in Fig A and Table A in S1 Supplementary Material.

**Simulating agents with selective belief distortions.**   To assess if certain belief distortions were sufficient to induce compulsions, we added a second set of simulations where only a single belief (agent parameter) was varied while the remaining beliefs were veridical (agent parameters were identical with underlying world parameters).

Specifically, we simulated 5,000 additional agents with the following constraints:

**Simulation 2A (partial, success belief):** Simulating agents with a pathological doubt about the success of their preventive action (handwashing): All agents had a veridical representation of the world, only their belief about successful washing ($p_{SUCCESS}'$) could vary ($\Delta(p_{SUCCESS})$ = *full range; $\Delta$ (all other parameters) = 0).*

**Simulation 2B (partial, threat magnitude):** Simulating agents with an overestimation of threat: All agents had a veridical representation of the world, only their belief about threat magnitude ($cost_{SICKNESS}'$) could vary ($\Delta(cost_{SICKNESS}')$ = *full range ; $\Delta$ (all other parameters) = 0).*

**Simulation 2C (partial, threat probability):** Simulating agents with an overestimation of threat: All agents had a veridical representation of the world, only their belief about threat probability ($p_{SOILING}'$) could vary ($\Delta(p_{SOILING}')$ = *full range; $\Delta$ (all other parameters) = 0).*

## Data analysis

**Differences in beliefs between compulsive and non-compulsive agents.**   To assess how the two groups differed in their (mis-)representation of the environment, we compared the agent parameters of the compulsive and non-compulsive group in Simulation 1A and B with the respective world parameters ($\Delta$ *params = agent params–world params,* whereby agent parameters are indicated by an apostrophe, e.g. ($\Delta p_{SUCCESS} = p'_{SUCCESS} - p_{SUCCESS}$) (see Table 1).

**Regression analysis with symptom severity.**   Second, we were interested how different parameters would contribute to the expression of compulsive episodes. To do so, we used regression analysis in order to directly relate the extent to which agents over- or underestimated the world parameters $\Delta$ *params* to the length and frequency of compulsive episodes for both Simulations 1 and 2.

## Statistical analyses

Statistical analyses were performed using Matlab (Version 9.3.0.713579 (R2017b). Natick, Massachusetts: The MathWorks Inc.). Unless stated otherwise, we used one sample *t*-tests for tests against 0 (function *ttest*) and two-sample *t*-test (function *ttest2*) for group comparisons. When the data did not follow a Normal distribution (bootstrapped Shapiro-Wilk test: p<0.05), we used a Wilcoxon-Mann-Whitney U-test (function *ranktest*), as always explicitly stated. Finally, we used Spearman tests for the correlation analysis. The correlation coefficient is indicated by Spearman's ρ. In order to avoid overinflated significance given our large sample size, all statistical tests were performed using a bootstrap method as follows: 1) agents were

randomly selected from the targeted groups (n = 20 for one sample tests, n = 40 for two sample tests (2 x 20) and correlations, to ensure a statistical power of 95% to detect large (d = 0.8) and very large (d = 1.2) effects respectively), 2) the usual statistical procedure was applied to this group, 3) the first two steps were repeated 10,000 times, 4) the mean (median for non-parametric tests) was computed to derive the sufficient statistics and p-values for the group. A p-value of $p < 0.05$ was considered significant.

## Supporting information

**S1 Supplementary Material. Fig A.** Results from Simulation 1B, otherwise Identical to Fig 3. **Table A.** Difference in world parameters and subjective beliefs (agent parameters), $\Delta$ params, for the compulsive and non-compulsive group in Simulation 1B (Mean (std)). Statistical comparison against 0 (no belief distortion) and between compulsive and non-compulsive group: bootstrapped t-test, d = Cohen's d. * indicates significance. **Table B.** Regression between compulsion severity and the degree of belief distortion for doubt about washing effectiveness $\Delta\, p_{SUCCESS}$, overestimation of threat magnitude $\Delta\, cost_{SICKNESS}$, overestimation of threat probability $\Delta\, p_{SOILING}$ and underestimation of correct observations $\Delta\, p_{DETECT\ DIRTY}$ and $\Delta\, p_{DETECT\ CLEAN}$ based on Simulations 1A (full simulation of all parameters). * indicates significance. $\rho$ is Spearman $\rho$. **Table C.** Regression between various aspects of compulsive episodes and the degree of belief distortion for washing effectiveness $\Delta\, p_{SUCCESS}$, overestimation of threat magnitude $\Delta\, cost_{SICKNESS}$ and overestimation of threat probability $\Delta\, p_{SOILING}$ based on Simulation 2A, B, C (selective belief distortions). * indicates significance. $\rho$ is Spearman $\rho$. **Table D.** Average belief (subjective probability of being in a dirty state) when performing each action in Compulsive and Non-compulsive agents based on Simulation 1A (full simulation of all parameters). Mean (Std). P-value from Bootstrapped U-test, d = Cohen's d. **Table E.** Action transition probabilities for Compulsive and Non-compulsive agents based on Simulation 1A (full simulation of all parameters). Mean (std). P-value from Bootstrapped U-test, d = Cohen's d. **Table F.** Absolute subjective beliefs (agent parameters) for the agents exhibiting pure checking (but no pure washing) compulsions, or pure washing (but no pure checking) in Simulation 1A and 1B (Mean (std)). Statistical comparison the pure checking and pure washing agents: Cohen's d and bootstrapped t-test. * indicates significance.
(DOCX)

## Author Contributions

**Conceptualization:** Lionel Rigoux, Klaas E. Stephan, Frederike H. Petzschner.

**Formal analysis:** Lionel Rigoux, Frederike H. Petzschner.

**Funding acquisition:** Klaas E. Stephan.

**Investigation:** Lionel Rigoux, Klaas E. Stephan, Frederike H. Petzschner.

**Methodology:** Lionel Rigoux, Frederike H. Petzschner.

**Project administration:** Klaas E. Stephan, Frederike H. Petzschner.

**Software:** Lionel Rigoux, Frederike H. Petzschner.

**Supervision:** Klaas E. Stephan, Frederike H. Petzschner.

**Validation:** Lionel Rigoux, Frederike H. Petzschner.

**Visualization:** Lionel Rigoux, Frederike H. Petzschner.

**Writing – original draft:** Lionel Rigoux, Klaas E. Stephan, Frederike H. Petzschner.

**Writing – review & editing:** Lionel Rigoux, Klaas E. Stephan, Frederike H. Petzschner.

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
