## [Decision Letter · Decision Letter 0]

16 Mar 2024

Dear Dr Petzschner, 

Thank you very much for submitting your manuscript "Beliefs, compulsive behavior and reduced confidence in control" for consideration at PLOS Computational Biology.

As with all papers reviewed by the journal, your manuscript was reviewed by members of the editorial board and by several independent reviewers. In light of the reviews (below this email), we would like to invite the resubmission of a significantly-revised version that takes into account the reviewers' comments.

We cannot make any decision about publication until we have seen the revised manuscript and your response to the reviewers' comments. Your revised manuscript is also likely to be sent to reviewers for further evaluation.

[1] A letter containing a detailed list of your responses to the review comments and a description of the changes you have made in the manuscript (we encourage your to follow these guidelines https://osf.io/preprints/psyarxiv/kyfus) . Please note while forming your response, if your article is accepted, you may have the opportunity to make the peer review history publicly available. The record will include editor decision letters (with reviews) and your responses to reviewer comments. If eligible, we will contact you to opt in or out.

Sincerely,

Stefano Palminteri

Academic Editor

PLOS Computational Biology

Lyle Graham

Section Editor

PLOS Computational Biology

Reviewer's Responses to Questions

**Comments to the Authors:**

Reviewer #1: This is a very interesting theoretical paper that shows that many aspects of OCD can be explained by a single parameter change in a cognitive computational model. The authors use a POMDP model of a cooking/handwashing/checking cleanliness scenario and many different simulations to investigations which parameters of the model might lead to compulsive behaviours (defined at both low and high thresholds), in scenarios in which the optimal behaviour did not include compulsions (due to the chosen parameters describing probabilities in the environment). They found that

- the only belief that distinguished compulsive from non-compulsive agents was a lower p(success) of handwashing than the actual p(success), and not other well-established hypotheses such as overestimation of threat magnitude/probability, or indeed the reliability of checking (or checking at all)

- the p(success) belief also affected the number, length and frequency of handwashing episodes

- although threat overestimation could trigger the onset of compulsive handwashing, only low p(success) was necessary and sufficient to sustain compulsions

- this core deficit then leads to an intolerance of uncertainty and perfectionism (e.g. in being extremely confident hands are clean before cooking)

- the relative costs of checking vs washing determine the proportions of these behaviours in the compulsions, however

The work is thorough and well-written and I support its publication - I have no major criticisms. Some relatively minor points are below:

In Figure 5C, I thought that the compulsive/non-compulsive groups were defined according to whether washing/checking actions were repeated without cooking in between. If so, then aren't the differences in transition probabilities illustrated here there by definition? It seems odd to do statistical testing on the criteria for selection to one group or another? It would make more sense to contrast groups with and without disparities between the believed and actual p(success).

p24 - it doesn't make sense to me to say "It implies that compulsive agents are less frequently exposed to the consequences of their washing actions". Compulsive agents are much more likely to check their hands are clean after washing than non-compulsive ones? This is surely exposure to the consequences of washing?

p38 - The authors draw a distinction between their proposal and the recent paper by Fradkin et al, saying that the latter merely propose greater uncertainty over all state transitions, whereas they more specifically propose that compulsions result from state-action transitions relating to harm prevention. I'm not sure these proposals are really so far from each other, however, as in the Fradkin paper those authors discuss action-dependent transitions in particular in the context of compulsion formation, and in this paper (on the next page) the authors cite experimental work showing that OCD patients are impaired learning about transitions of seasons (which are presumably not action-dependent?). So really it is the balance of general vs specific deficits that tilts slightly different ways in these papers. To me the biggest difference between the papers is the robustness of the analysis here: they have done thousands of simulations demonstrating both sufficiency and necessity of the action-dependent belief in causing compulsions (and excluding other potentially responsible parameters).

p41 - The authors comment on the elevated sense of responsibility in OCD but say their model stops short of explaining this phenomenon. Reading this I wondered whether an asymmetry in the belief about the efficacy of preventative actions (i.e. doubting actions that prevent harm more than those that cause harm) does in fact account for this? After all, we all have urges to do things that we then terminate at some stage of planning/execution. Most of us can be confident that a) we could perform such an action, and b) we could stop ourselves from performing the action. If these can be regarded as separate actions, then for an OCD patient, presumably just imaging themselves doing something bad would inevitably make that action more likely, as they believe they are less able to prevent it than to execute it? This must lead to a great sense of responsibility over what one even contemplates. The authors are under no obligation to include these speculations however!

Reviewer #2: In this paper, Rigoux et al, propose a quantitative modeling approach of Obsessional-Compulsive Disorder (OCD), by simulating the behavior of human agents described by a limited set of parameters characterizing their beliefs formation process in a simplified paradigmatic symptom-triggering situation (handwashing) which the agents are repeatedly exposed to, under a normative learning decision scheme (Partially Observable Markov Decision Process, POMDP) that dictates their evolution. By systematically varying agents' parameters' values, Rigoux et al observe that the critical trait of agents exhibiting compulsive behaviors in this scenario (and the main factor associated with the intensity of such compulsive behaviors) was the underestimation of the effectiveness of one's threat-eliminating action (here cleaning) in contrast to other dysfunctional beliefs pertaining to, for example, overestimating the danger associated with dirt, being hypersensitive to dirt, etc. all of which are hypothesized core dysfunctional beliefs according to the most prevalent psychological theories of OCD. In addition, having such dysfunctional beliefs about the outcome of one's action nurtured a less efficient belief updating in the simulated agents, which authors associate with the exacerbated uncertainty and pathological doubt self-reported by OCD patients.

This computational psychiatry simulation study represent a very welcome endeavor to confront multiple hypotheses regarding the cognitive roots of OCD focussing on the dysfunctional beliefs that may operate in the decision-making process of patients. Empirical studies fail to arbitrate between these theories which make a number of common predictions, though not quantitative ones. That is where a modeling work like this one represents an insightful progress by directly comparing the consequences of each of these hypothetical core dysfunctional beliefs as they are implemented in a population of evolving simulated agents under the exact same circumstances. My opinion is that more work of that quality should be published as it would help moving the field of cognitive psychopathology forward.

Obviously, the numerical tracking of these agents requires the authors to restrain themselves to a simplified scenario and to assume each agent to obey optimal decision making under uncertainty. And this may feel like oversimplification to the clinically-minded reader. Yet the authors make a great effort to translate, ex ante, how clinical hypotheses might be subsumed in specific design choice of their model and, ex post, the meaning of the outcome of the simulations. Yet I feel that on critical aspect that may escape the reader unfamiliar with such modeling work is that the modeled agents are plunged in an iterated (or sequential) learning framework under which they adjust their beliefs about the world through repeated instances of actions and outcomes.

The authors also suggest a number of empirical predictions that could be derived from their in-silico results. This is also very much commendable as this effort is only too rarely made by authors in computational works, including computational psychiatry research.

Having little to critique on the heart of the simulation work, I feel that the discussion should expand on an important though not critical issue: As acknowledged by the author themself, the agent model they deploy does not allow the parameter to change and adjust (e.g. as could be achieved in a Bayseian framework as suggested by the authors themselves). As a consequence, their model can not inform on the onset of OCD while a number of clinical presentations seem to appear in the young adults with little if any earlier signs or symptoms. In my opinion the other should discuss how their modeling work speaks to a developmental / trait disorder and how to interpret life-long learning with respect to their sequential framework.

Finally, the authors should be commended for making all code, data and intermediate computing state, down to figure rendering freely and easily available on their github. However, I must say the high-performance computer resources required to re-run the analyses limit my ability to actually track the full extent of the data.

Reviewer #3: This is a very interesting paper, I very much enjoyed reviewing this. The paper uses a computational modelling approach, based on POMDPs to identify through simulations what could cause compulsive and obsessive symptoms in OCD. They find both expected (a faulty believe in how likely handwashing is to clean hands leads to compulsive behaviour) and novel results (beyond these believe abnormalities, no further changes needs to be made to the effectiveness of checking to produce compulsive behaviour). Interestingly, they can also use their model to explain previously reported findings (learning deficits in OCD) that were not explicitly included in the model. The code accompanying the paper is freely available and annotated.

I only two one ‘major’ comment (but they are pretty small points of criticism actually):

1. Can your model also be fit to agents to test parameter recovery? If not, how would you use your paradigm in a study of actual participants? (You say in the online code annotations that some of it took weeks to run on a cluster) E.g. if you had data from real participants in a task, would you be able to estimate whether their behaviour is due to uncertainty about washing success and checking success? Please could you say something about this in the discussion.

2. It is intriguing that (figure 3), compulsive agents differ in their estimation of being able to successfully clean, rather than e.g. their ability to check. Your discussion of this and relating it to the phenomenology of ‘not-just-right’ is very interesting. Do you find that when analysing the behaviour of these agents that they do have also an increase of check actions, or do they just differ in their washing actions? Would it be possible to somehow illustrate the raw behaviour, e.g. something similar to figure 2B but across the different agents? This is somewhat addressed in figure 4A, but if I understand correctly, this is not separate for episodes defined based on checking or on washing? Somewhat conceptually related, in recent data in a large sample by Trier et al. (2023, preprint), there was no difference for checking actions with a transdiagnostic compulsivity dimension, but there were changes to active avoidance behaviour. (However, that data were not modelled with POMDP, so it is not directly comparable unfortunately).

Here are my minor comments, almost all of these refer to specific places in writing (mainly methods) where some clarity could be added:

3. The authors put their code online. The code is very tidy and well annotated inside the code. However, if there was something like a tutorial walk through, I wonder whether this would make this approach easier for other people to implement in their studies? E.g. something that would link the specific aspects described in the methods to the code.

4. In the methods you say on page 47 “In principle, this can be numerically computed, using for example, dynamic programming.” Just to clarify, do you mean that you did this using dynamic programming? Or are you saying you could have done that, but you did something else?

5. Reading table 4, I did not quite realize what is a parameter. And I also did not quite understand what is meant by parameter – e.g. in a decision making task it would be ‘sensitivity to reward’. Could I clarify that here then parameter would mean e.g. for cost_sickness ‘how sensitive the agent is to the cost of sickness’ – I think I got confused because I assumed that this was determined by the environment. Maybe please clarify what is a characteristic of the environment – and thus set by the experimenter – and what is instead a property of how the agent perceives the environment.

6. In table 5, I don’t understand why the probability for observing ‘clean’ is 50% when cooking or washing. I would have assumed that – in the absence of checking – this would instead be a free parameter, something like capturing your belief that because you have cooked, it must be dirty now (maybe combined with a prior belief about whether your hands were already dirty). Or no update to the believes at all in the absence of a check action. I don’t quite understand what the 50% is meant to capture instead.

7. On page 55, when talking about optimal policy, please could you clarify whether the policy was optimal given full knowledge of the world? Or given subject parameters that mapped correctly onto the world, or what exactly was optimal – e.g. maximum rewards achieved empirically?

<b>Have the authors made all data and (if applicable) computational code underlyi

---

## [Decision Letter · Decision Letter 1]

28 May 2024

Dear Petzschner,

We are pleased to inform you that your manuscript 'Beliefs, compulsive behavior and reduced confidence in control' has been provisionally accepted for publication in PLOS Computational Biology.

Best regards,

Stefano Palminteri

Academic Editor

PLOS Computational Biology

Lyle Graham

Section Editor

PLOS Computational Biology

Reviewer's Responses to Questions

**Comments to the Authors:**

Reviewer #1: Thank you for responding to my comments - I hope they were helpful. I have no further points.

Reviewer #3: The authors have addressed all of my questions.

**Have the authors made all data and (if applicable) computational code underlying the findings in their manuscript fully available?**

Reviewer #1: Yes

Reviewer #3: Yes

PLOS authors have the option to publish the peer review history of their article (what does this mean?). If published, this will include your full peer review and any attached files.

Reviewer #1: No

Reviewer #3: No

---

## [Editor Report · Acceptance letter]

11 Jun 2024

PCOMPBIOL-D-24-00019R1 

Beliefs, compulsive behavior and reduced confidence in control

Dear Dr Petzschner,

I am pleased to inform you that your manuscript has been formally accepted for publication in PLOS Computational Biology. Your manuscript is now with our production department and you will be notified of the publication date in due course.

With kind regards,

Anita Estes
